# *Mycobacterium tuberculosis* exploits host ATM kinase for survival advantage through SecA2 secretome

Savita Lochab[1,2], Yogendra Singh[2], Sagar Sengupta[1], Vinay Kumar Nandicoori[1]*

[1]National Institute of Immunology, New Delhi, India; [2]Department of Zoology, University of Delhi, New Delhi, India

**Abstract** (*Mtb*) produces inflections in the host signaling networks to create a favorable milieu for survival. The virulent *Mtb* strain, *Rv* caused double strand breaks (DSBs), whereas the non-virulent *Ra* strain triggered single-stranded DNA generation. The effectors secreted by SecA2 pathway were essential and adequate for the genesis of DSBs. Accumulation of DSBs mediated through *Rv* activates ATM-Chk2 pathway of DNA damage response (DDR) signaling, resulting in altered cell cycle. Instead of the classical ATM-Chk2 DDR, *Mtb* gains survival advantage through ATM-Akt signaling cascade. Notably, in vivo infection with *Mtb* led to sustained DSBs and ATM activation during chronic phase of tuberculosis. Addition of ATM inhibitor enhances isoniazid mediated *Mtb* clearance in macrophages as well as in murine infection model, suggesting its utility for host directed adjunct therapy. Collectively, data suggests that DSBs inflicted by SecA2 secretome of *Mtb* provides survival niche through activation of ATM kinase.

## Introduction

In response to the damage, the host activates an intricate and indispensable signaling cascade entitled '**D**NA **d**amage **r**esponse' (DDR), which not only detects and repairs the damaged lesions in DNA but also regulates the activation of effectors that determine the fate of the cell. Ataxia telangiectasia mutated (ATM), ATM- and Rad3-related protein (ATR) and DNA-dependent protein kinase catalytic subunit (DNA-PKcs) are three drivers of DDR which belong to the family of phosphoinositide 3-kinase like kinases (PIKKs). PIKKs respond to the DNA damage by activating their downstream substrates leading to cell cycle delay/arrest and DNA repair and depending on the extent of damage can also lead to apoptosis. ATM is activated through autophosphorylation of S1981 residue and is subsequently recruited to the double strand breaks (DSBs) through a sensor complex, MRN (MRE11, NBS1 and Rad50) (*Lee and Paull, 2005*). DSBs are also the site of recruitment and activation of DNA-PKcs, facilitated by a DSB-bound heterodimer Ku70-80 (*Gottlieb and Jackson, 1993*; *Spagnolo et al., 2006*). ATR-ATRIP complex is recruited in response to RPA-coated ssDNA (*Ball et al., 2005*). Even though all the three kinases coordinate their functions independently, crosstalk among these kinases is known to exist, such as ATM-dependent activation of ATR (*Shiotani and Zou, 2009*); ATM and ATR-dependent phosphorylation of DNA-PKcs (*Chen et al., 2007*; *Hirokawa et al., 1992*) and DNA-PKcs-mediated modulation of ATM (*Peng et al., 2005*; *Zhou et al., 2017*). Moreover, these kinases also share substrates such as p53 (*Canman et al., 1998*; *Lakin et al., 1999*; *Lees-Miller et al., 1990*). Phosphorylation of H2AX at the serine 139 (termed γH2AX), at the chromatin regions flanking the damage site is considered as a marker for DNA damage (*Burma et al., 2001*; *Rogakou et al., 1998*). While phosphorylation of H2AX is predominantly modulated by ATM, ATR and DNA-PKcs also contribute either partially or entirely (*Burma et al., 2001*; *Royo et al., 2013*; *Wang et al., 2005*). Phosphorylation of H2AX acts as the foundation for recruitment of other DDR mediator proteins eventually leading to DNA repair (*Ciccia and Elledge,*

*For correspondence:
vinaykn@nii.ac.in

**Competing interests:** The authors declare that no competing interests exist.

*2010*). Thus, DDR is an indispensable mechanism that restores the genomic integrity and regulates the cellular response by modulating DNA repair, cell cycle progression, apoptosis, or senescence in response to DSBs in the cell. Thus, DDR is also a preferred target of pathogenic bacteria.

Pathogenic *Escherichia coli* and *Helicobacter pylori* are capable of imparting notable DNA damage to the host and subsequently impair the DDR to avoid premature cell death (*Cuevas-Ramos et al., 2010*; *Toller et al., 2011*). *Chlamydia trachomatis* triggers DSBs in the host, and γH2AX induction but simultaneously it impairs the DDR by inhibiting the recruitment of 53BP1, ensuing inadequate signal amplification (*Chumduri et al., 2013*). Listeriolysin O (LLO) secreted by *Listeria monocytogenes* induces degradation of crucial DNA damage sensor, MRE11. This results in impaired DDR, curtailing the host ability to halt cell cycle, thus successfully promoting multiplication and survival of the pathogen (*Samba-Louaka et al., 2014*). *H. pylori* impedes nucleotide repair by downregulating the proteins and their respective transcripts that are involved in mismatch and base excision repair (*Kim et al., 2002*; *Machado et al., 2009*). *Neisseria gonorrhoeae*, *H. pylori* and *C. trachomatis* also downregulate p53 levels to promote host cell survival and inhibit apoptosis (*Buti et al., 2011*; *Wei et al., 2010*; *Vielfort et al., 2013*).

Since ancient times *Mycobacterium tuberculosis* (*Mtb*) has been steadily evolving sophisticated tactics to dodge the defense responses of the host. One of the remarkable survival strategies of *Mtb* is to intervene with the fundamental signaling events of the host cell (*Koul et al., 2004*) and to facilitate these manipulations *Mtb* secretes an enormous number of characterized and uncharacterized effectors inside the host. These effectors modulate host cellular processes such as phagosome maturation, apoptosis, autophagy, calcium homeostasis, activation of pro-inflammatory responses and TLR, TNFα, MAPK signaling pathways (*Dey and Bishai, 2014*). However, till date, the role of ATM kinase in the survival of *Mtb* inside the host has not been investigated.

In this study, we demonstrate that *Mtb* causes DSBs and determine its impact on the activation of host DDR. SecA2 secretome is necessary and sufficient for inflicting DSBs in the host. We show that instead of classical ATM-Chk2 pathway, *Mtb* gains survival advantage through activation of ATM-Akt signaling cascade that results in the inhibition of apoptosis. In a chronic mice infection model, *Mtb* infected lungs showed significant DSBs and activation of ATM. Combining ATM inhibitor, KU55933 with INH resulted in better clearance of *Mtb* compared with INH treatment alone in the lungs and spleen of infected mice. This study reveals novel exploitation mechanism utilized by *Mtb*, wherein the pathogen inflicts persistent DSBs in the host to activate ATM-Akt signaling pathway and thereby inhibiting apoptosis and accentuating cell growth.

## Results

### Mtb inflicts DNA damage in the host cell

To address the question of whether *Mtb* infection leads to the damage of the host DNA we used PMA differentiated THP-1, RAW264.7 (RAW) macrophages and primary murine peritoneal macrophages (PΦ). The cells were infected with the virulent *Mtb* strain, H37Rv (*Rv*) and the γH2AX levels, the hallmark of DNA damage, was evaluated. Results showed considerable DNA damage in the infected cells compared to the corresponding uninfected control and these observations were consistent across all the three cell types (*Figure 1a–c*). The damage could be observed as early as 1 hr post infection (p.i) (*Figure 1b*) and persisted even at 72 hr (*Figure 1a*). We assessed if the observed DNA damage is dependent on the presence of live bacteria by infecting cells with live or heat killed *Rv*. Results showed that (*Figure 1d*) only the live bacilli could cause damage to the host genome. To evaluate the role of *Mtb* virulence in inflicting genotoxicity, we performed infection experiments with *Rv* or its avirulent counterpart H37Ra (*Ra*). Based on its characteristics such as reduced survival under anaerobic conditions, inability to produce persistent infection in mice and guinea pigs, *Ra* is considered as attenuated, avirulent strain of *Mtb* (*Alsaadi and Smith, 1973*; *North and Izzo, 1993*). While both *Ra* and *Rv* infected THP1 cells showed considerably higher γH2AX levels compared with the uninfected control, there was consistent and noticeably higher levels of γH2AX levels in *Rv* compared with the *Ra* infected cells (*Figure 1e*; compare 4, 24 and 48 hr time points). However, in case of RAW cells the higher γH2AX levels in *Rv* was apparent only in the early time points (*Figure 1f and g*: up to 4 hr). Subsequent to DNA damage, H2AX gets phosphorylated on the chromatin flanking the site of damage, which appears as foci in the nucleus. We performed immunofluorescence

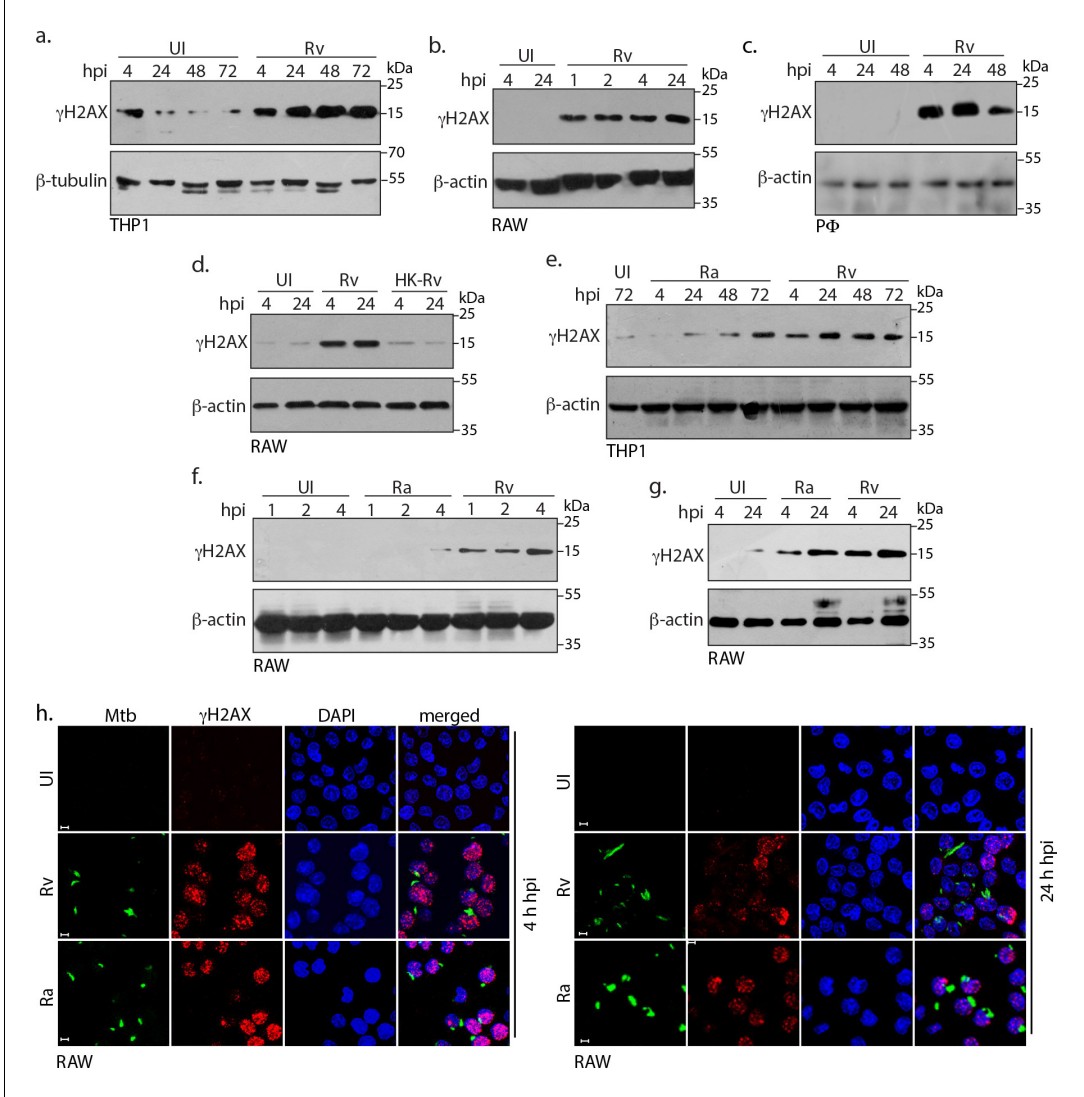

**Figure 1.** Mtb inflicts DNA damage in the host cell. (a) PMA differentiated THP-1 cells were infected with *Rv* for 4 hr. Cells were washed thrice with 1XPBS to remove extracellular bacilli and supplemented with fresh media for further time points which were calculated with respect to 4 hr point. (b and c) RAW264.7 macrophages (RAW) (b) and murine peritoneal macrophages (PΦ) (c) were infected with *Rv* as described in (a) except that the extracellular bacilli were washed off after 1 and 2 hr post-infection (p.i) For the remaining time points, 4 hr serves as the reference point. (d) RAW264.7 macrophages were infected with *Rv* or heat-killed *Rv* (HK-*Rv*) and cell lysates were prepared at 4 and 24 hr p.i (e–g) THP-1 or RAW264.7 cells were infected with *Rv* and *Ra*. Cell lysates were prepared as indicated above after 1, 2, 4, 24, 48 or 72 hr p.i. (a–g) Whole cell lysates (WCL) were resolved on SDS-PAGE, transferred to nitrocellulose membrane and probed with α-γH2AX(S139) and α-β-actin/α-β-tubulin antibodies. (h) Representative immunofluorescence image showing γH2AX foci (AlexaFluor594-Red) and nucleus (DAPI-blue) in RAW264.7 infected with GFP expressing *Rv* and *Ra* (Green) independently at 4 and 24 hr p.i. Images were captured at 63X magnification with scale bar of 5 μm.

The online version of this article includes the following figure supplement(s) for figure 1:

**Figure supplement 1.** Mtb inflicts DNA damage in the host cell.

experiments using antibodies specific for γH2AX and observed significantly higher numbers of γH2AX foci at both 4 and 24 hr p.i (*Figure 1h*) compared with the uninfected controls. Consistent results were observed in infected THP1 and PΦ cells (*Figure 1—figure supplement 1a and b*). Collectively, the data suggests that both *Rv* and *Ra* possess genotoxic characteristics and virulent *Rv* strain triggers rapid damage post infection.

## Host cell activates DNA damage response

Subsequent to the challenge to genome integrity, eukaryotic cells activate the DDR, which initiates either cell survival or cell death signals contingent upon the magnitude of damage. Depending on the type of damage, - DSBs or single-stranded DNA (ssDNA; due to replication fork stalling) either ATM-Chk2 or ATR-Chk1 pathways, respectively, are activated (*Figure 2a*; *Lee and Paull, 2007*; *Zou and Elledge, 2003*). To determine the type of DDR involved in *Ra* and *Rv* inflicted damage, lysates prepared at 4 and 24 hr p.i were examined for the phosphorylated forms of ATM, Chk2, ATR and Chk1, indicative of their activation. Infection by *Ra* led to the formation of distinct nuclear pRPA2 (*Figure 2c*) and RPA2 (*Figure 2—figure supplement 1a*) foci and robust activation of ATR-Chk1 pathway (*Figure 2b*; right panel), suggesting that *Ra* induces generation of ssDNA. On the other hand, infection by *Rv* led to significant activation of ATM-Chk2 pathway in both RAW and PΦ cells, suggesting occurrence of DSBs in the host nuclei (*Figure 2b*, left panel and 2d). These results were corroborated by immunofluorescence experiments, wherein we observed distinct pATM foci upon *Rv* infection (*Figure 2e*). Furthermore, we assessed the transmission of DNA damage signals by evaluating the expression and activation levels of sensors, transducers and mediators upon *Rv* infection. While there were no significant perturbation in the protein levels, enhanced phosphorylation of Nbs1, MDC1, 53BP1, representing the activated forms, was evident upon *Rv* infection (*Figure 2—figure supplement 1b and c*; *Shiloh, 2003*). To corroborate these findings, we evaluated γH2AX foci for the presence of mediator proteins 53BP1, which accumulates at the DNA damage site. Distinct focal pattern and colocalization of γH2AX and 53BP1 foci in *Rv* infected PΦ supported the argument that infection with *Rv* induces DSBs (*Figure 2f*).

## Intracellular Mtb causes continuous DSBs

*Mtb* infection not only elicits early DNA damage (*Figure 1f*) but that also persists at higher levels for prolonged duration (*Figure 1a, e* and *Figure 3a* left panel). DDR is necessary for sensing, amplifying and eventually repairing the damage. However, persistent DSBs indicate either inefficient DDR or continued presence of the factors released by *Rv* that are responsible for the damage. To determine whether there is a direct correlation between the levels of γH2AX and *Rv* load, we evaluated γH2AX levels in the absence or presence of isoniazid (INH), the front line anti-tuberculosis antibiotic (*Nuermberger and Grosset, 2004*). Analysis of colony-forming units (CFUs) at different time points post *Rv* infection ascertained efficient clearance of intracellular pathogen in the presence with INH (*Figure 3b* and *Figure 3—source data 1*). Presence or absence of INH did not influence the extent of γH2AX levels in uninfected control cells (*Figure 3a*; right panel). While γH2AX levels continued to persist in *Rv* infected cells, treatment of INH led to time dependent reduction in γH2AX, pATM and pChk2 levels (*Figure 3c*), concomitant with the clearance of the pathogen (*Figure 3b*). Thus, the data suggests that even though DDR is active, inefficient repair is due to continuous generation of DSBs during prolonged *Rv* infection.

Apart from activating signals for repair, the DDR stalls the DNA replication and cell division by activating the cell cycle checkpoints (*Smith et al., 2010*). To examine the consequence of *Rv*-induced DSBs on the host cell cycle progression, RAW cells were infected with GFP-expressing *Rv* and the populations of infected and uninfected cells were analyzed through flow cytometry (*Figure 3d*). Flow cytometry profile of uninfected control cells would show single population, whereas *Rv*-infected RAW cells would have two populations: GFP+ve cells harboring intracellular GFP-*Rv* and GFP-ve uninfected bystander cells that lack intracellular GFP-*Rv* (*Figure 3d*). Interestingly, compared with the uninfected cell population, *Rv* infected GFP+ve population showed decreased accumulation of cells in G1 phase, soon after the infection (4 h p.i). Lowered accumulation of cells in the G1 phase suggest alterations in the progression of cell cycle in the cells harboring intracellular bacilli (*Figure 3e*). Importantly, at late stages of infection we observed concomitant increase in the sub-G1 cell population, indicative of cell death (*Figure 3e*). DDR activation maintains the genomic stability by delaying the occurrence of cell division till DNA is repaired, else in case of irreversible damages; key regulators such as p53 mediates apoptosis (*Figure 3f*). To corroborate flow cytometry results, we evaluated key molecular markers that modulate the transition of cells through different cell cycle phases (*Abraham, 2001*; *Matsuoka et al., 1998*). Elevated levels of p53 and p21 upon *Rv* infection are reflective of molecular level alterations in the cell cycle leading to

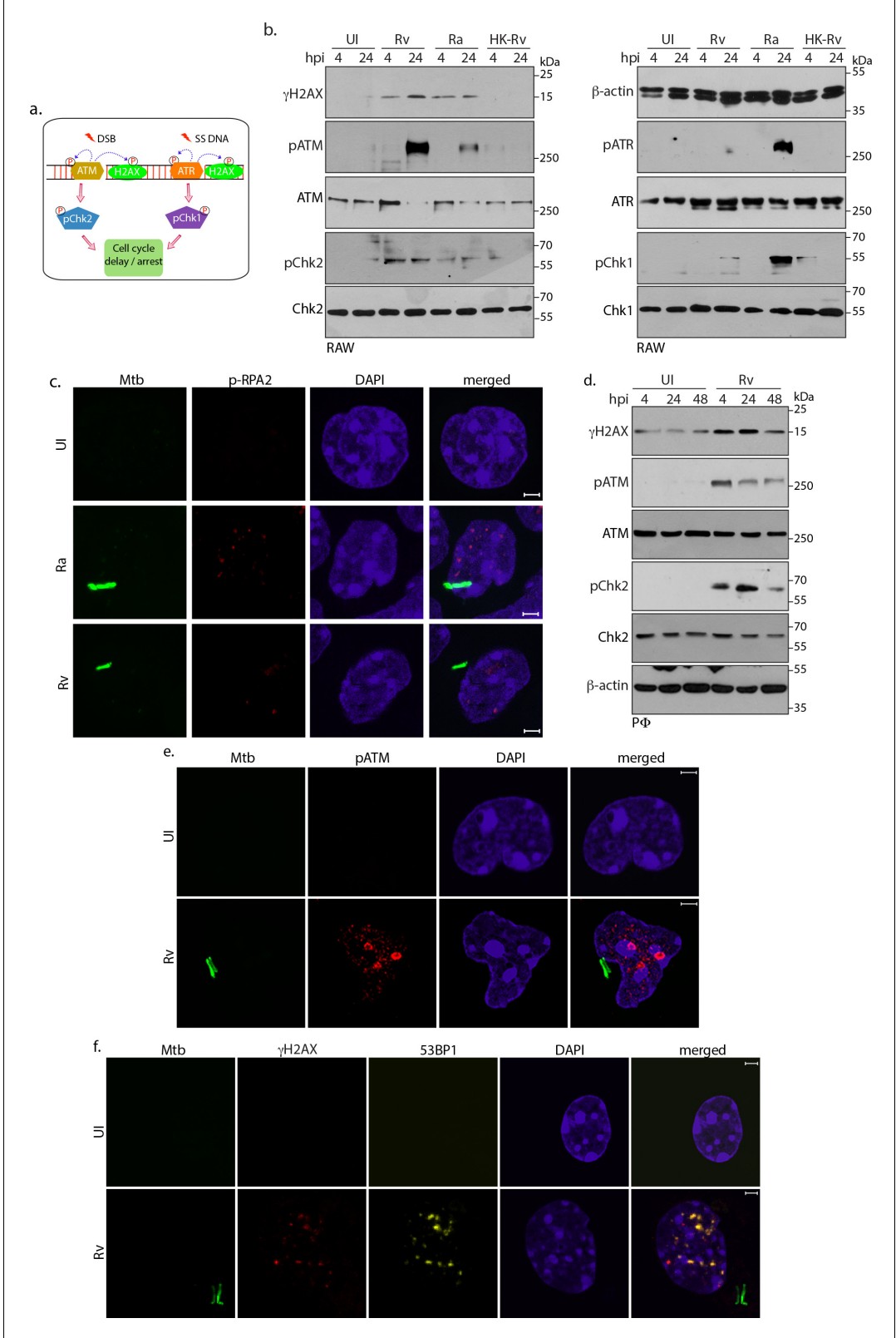

**Figure 2.** Host cell activates DNA damage response. (**a**) Schematic outline of ATM-Chk2 and ATR-Chk1 DDRs. Double strand breaks (DSBs) mediate the activation of ATM and its corresponding effector, Chk2 whereas ssDNA originated due to replication stalling leads to the activation of ATR and the downstream effector Chk1. Chk2 and Chk1 further activate cell cycle checkpoints to delay or arrest cell cycle progression. Cells with extensive DNA damage exit the route of cell cycle and undergo apoptosis. Phosphorylation status of proteins is depicted with P in a circle (**b**) To determine the

*Figure 2 continued on next page*

*Figure 2 continued*

signalling cascade coordinating the DNA damage, RAW264.7 macrophages infected with *Rv, Ra* or *HK-Rv*. WCL prepared at 4 and 24 h p.i. were subjected to immunoblotting with α-γH2AX, α-pATM(S1981), α-ATM, α-pCHK2(T68), α-CHK2 (left panel) and α-β-actin, α-ATR(S428), α-ATR, α-pCHK1 (S345), α-CHK1 (right panel) antibodies. (**c**) Representative immunofluorescence showing presence of pRPA2-S4/8 foci (AlexaFluor594-Red) in uninfected or GFP-*Rv*/GFP *Ra* (Green) infected RAW264.7 macrophages at 24 hr (**d**) WCL prepared from PΦ infected with *Rv* for 4 and 24 hr were immunoblotted with α-γH2AX, α- pATM(S1981), α-ATM, α-pCHK2(T68), α-CHK2 and α-β-actin antibodies. (**e–f**). Representative images of immunofluorescence showing (**e**) pATM(S1981) (AlexaFluor594-Red) and (**f**) γH2AX, 53BP1 foci formation (AlexaFluor594-Red and AlexaFluor647-yellow) in PΦ infected with GFP expressing *Rv* (Green) for 24 hr. Nuclei were stained with DAPI. Images were captured at 100X magnification with 2.9 X optical zoom in LSM510 Meta System (Zeiss, Germany) confocal microscope. Scale bar: 2 μm.

The online version of this article includes the following figure supplement(s) for figure 2:

**Figure supplement 1.** Host cell activates DNA damage response.

eventual cell death indicated as sub-G1 population (*Figure 3g*). Taken together, presence of *Rv* inside host causes constant DNA damages followed by cell cycle perturbations.

## Mtb SecA2 secretome is necessary and sufficient for DNA damage

Pathogenic bacteria often release genotoxins and cyclomodulins to modulate host cellular processes (*Nougayrède et al., 2005*). Whi, WhiB3, a redox regulator in *Mtb* has been shown to regulate the production of specific polyketides and lipid cyclomodulins in the host (*Cumming et al., 2017*). We speculated that the *Mtb* secretome might play a role in promoting DNA damage. Culture filtrate (CF) of *Mtb* contain proteins that are translocated into the extracellular milieu (*Figure 4a*). Addition of *Mtb* CF resulted in robust γH2AX induction, even higher than those observed upon *Rv* infection (*Figure 4b*). Importantly as little as 1 μg/ml CF treatment induced significant γH2AX (*Figure 4c*). In *Mtb,* proteins are secreted using both classical secretory pathways such as TAT and SecA1, which are essential for in vitro growth; and accessory secretion pathways such as SecA2 and Type VII secretion systems, which are necessary for survival and virulence in the host (*Feltcher et al., 2010*; *Ligon et al., 2012*; *Miller et al., 2017*). The TypeVII secretion system and its effectors are encoded by Region of difference 1 (RD1) in *Mtb*. Host cells infected with *Rv* or *RvΔRD1(Rv* carrying RD1 deletion) or RvΔCE (*Rv* strain with CFP10-ESAT6 deletion) had comparable γH2AX levels (*Figure 4d & f*). On the other hand, expression of γH2AX observed upon infection with *RvΔsecA2* (SecA2 exporter mutant) was significantly lower at the later time points (*Figure 4e & f*). However, since the deletion of *secA2* compromises the survival of *Mtb* in the host (*Braunstein et al., 2003*; *Kurtz et al., 2006*; *Miller et al., 2017*), we were unable to rule out the possibility that the decreased γH2AX levels could be due to better clearance of the pathogen. To address this issue, cells were treated with CFs prepared from either *Rv* or *RvΔRD1* or *RvΔsecA2* and the γH2AX levels were evaluated. While the γH2AX levels in cells treated with CF from *RvΔRD1* were comparable to those observed upon *Rv*-CF treatment; cells treated with *RvΔsecA2*-CF showed negligible γH2AX levels (*Figure 4g*). Treatment of cells either with *Rv-CF or RvΔRD1-CF* led to the activation of DDR players, ATM and Chk2. However, treatment with *RvΔsecA2-CF* cells did not show ATM-Chk2 activation. Analogous results were obtained when the experiment was performed with PΦ (*Figure 4h*). Together, results suggest that effectors secreted by SecA2 pathway are necessary and sufficient for inflicting genotoxic stress in the host.

## ATM activation confers survival advantages to Mtb

DNA damage mediates the activation of ATM kinase, which in addition to activating DDR is also known to modulate other cellular processes such as apoptosis, DNA repair, cell division, autophagy and inflammatory responses (*Kastan and Lim, 2000*). We assessed the impact of host ATM kinase activation in regulating the survival of *Mtb* with the help of KU55933 (ATM-I), a specific inhibitor of ATM activation (*Hickson et al., 2004*). Treatment of infected cells with ATM-I inhibitor resulted in substantial decrease in the γH2AX levels and abrogation of ATM and Chk2 activation (*Figure 5a*). This was also reflected in disappearance of pATM foci upon ATM-I treatment in *Rv*-infected cells (*Figure 5b*). Notably, ATM-I treatment of *Rv*-infected cells resulted in reduced survival of the pathogen in a dose dependent manner (*Figure 5c* and *Figure 5—source data 1*). However, ATM-I did not impact in vitro Mtb growth suggesting that the clearance of the bacilli is mediated through inhibition of ATM activation (*Figure 5—figure supplement 1*). Next, we examined if the compromised *Mtb*

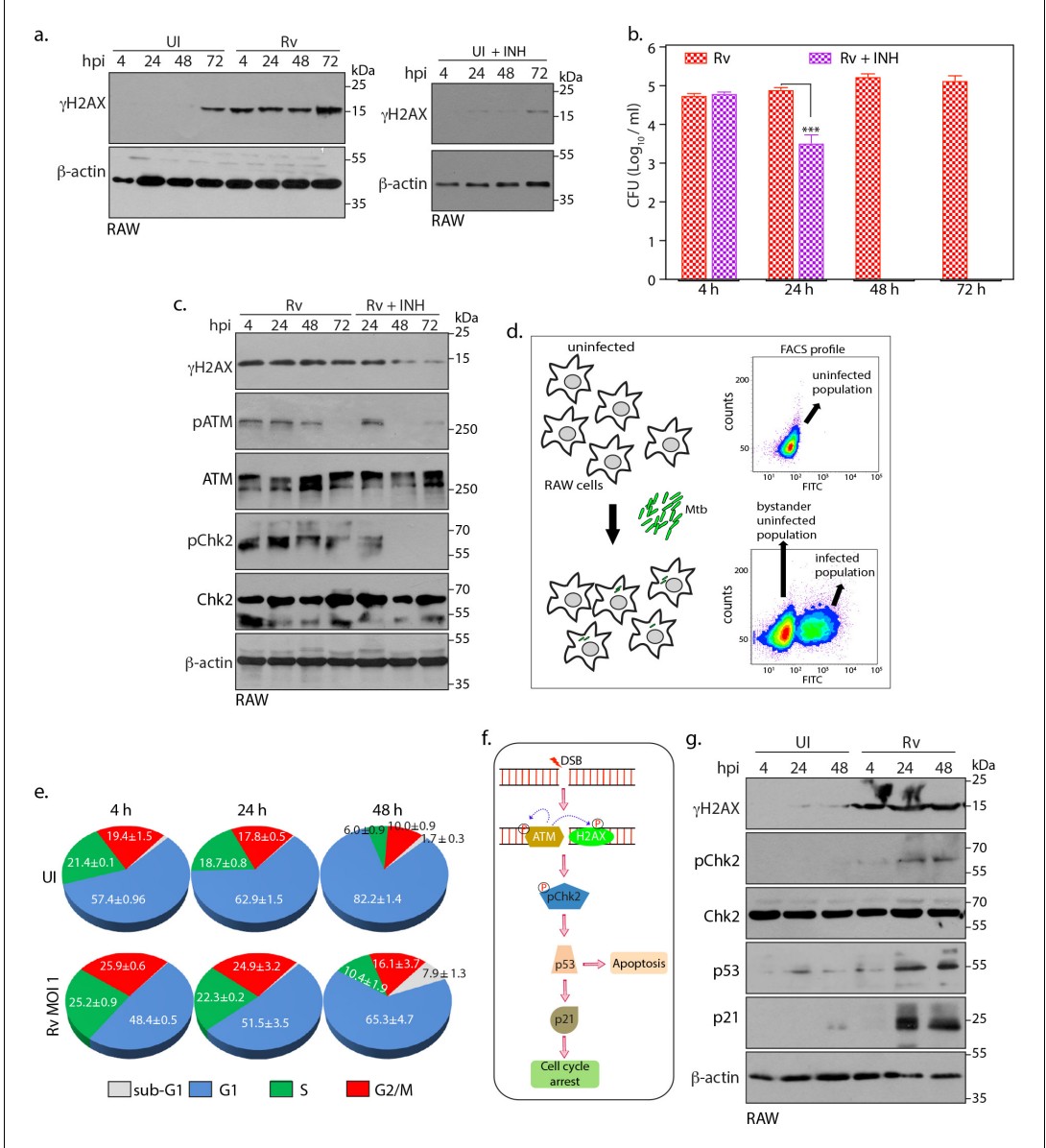

**Figure 3.** Intracellular Mtb causes continuous DSBs. (a) Left panel: RAW264.7 cells were infected with *Rv*. Cells were harvested at indicated time points to prepare WCLs which were then probed with α-γH2AX and α-β-actin antibodies. Right panel: RAW cells were treated with 1 μg/mL (7.3 μM) INH for indicated time points. WCL were prepared and immunoblotted with α-γH2AX and α-β-actin. (b and c) RAW264.7 cells were infected with *Rv*. After 4 hr, extracellular *Rv* were washed with PBS and supplemented with fresh media with or without 7.3 μM isoniazid (INH) (b). Cells at indicated time points were lysed with 0.05% SDS, serially diluted and plated on 7H11 plates to enumerate bacillary load. Bar graph on *y-axis* represents Mean CFU (log₁₀) ± SD of one of the three independent experiments performed in triplicates. **p≤0.005, ***p≤0.0005, (c). WCL were prepared at indicated time points were probed with α-γH2AX, α-pATM(S1981), α-ATM, α-pChk2(T68), α-Chk2 and α-β-actin antibodies. (d) Schematic diagram illustrating the experimental approach for the cell cycle analysis using flow cytometry. RAW264.7 cells were infected with GFP expressing *Rv*. FACS profile showing GFP+ve population containing *Rv* and GFP-ve population that lacks intracellular *Rv* (bystander uninfected cells). Uninfected control cells are GFP-ve. (e) Pie chart showing percentage of uninfected and GFP+ve infected cells in different phases of cell cycle. Infection was performed at the indicated timepoints. (f) Schematic representation of signaling activated due to DSB occurrence in the host cell. ATM phosphorylates its self and H2AX. Subsequently it activates the downstream effector, Chk2 at T68. Activated Chk2 stabilizes p53 which in turn elevates p21 levels. Upregulation of p53 can divert cells toward apoptosis while p21 leads to cell cycle arrest (g). WCLs prepared from RAW264.7 infected with GFP expressing *Rv* for indicated time points were subjected to immunoblotting with α-γH2AX, α-p21, α-p53, α-pChk2(T68), α-Chk2 and α-β-actin antibodies.

The online version of this article includes the following source data for figure 3:

**Source data 1.** Intracellular Mtb causes continuous DSBs.

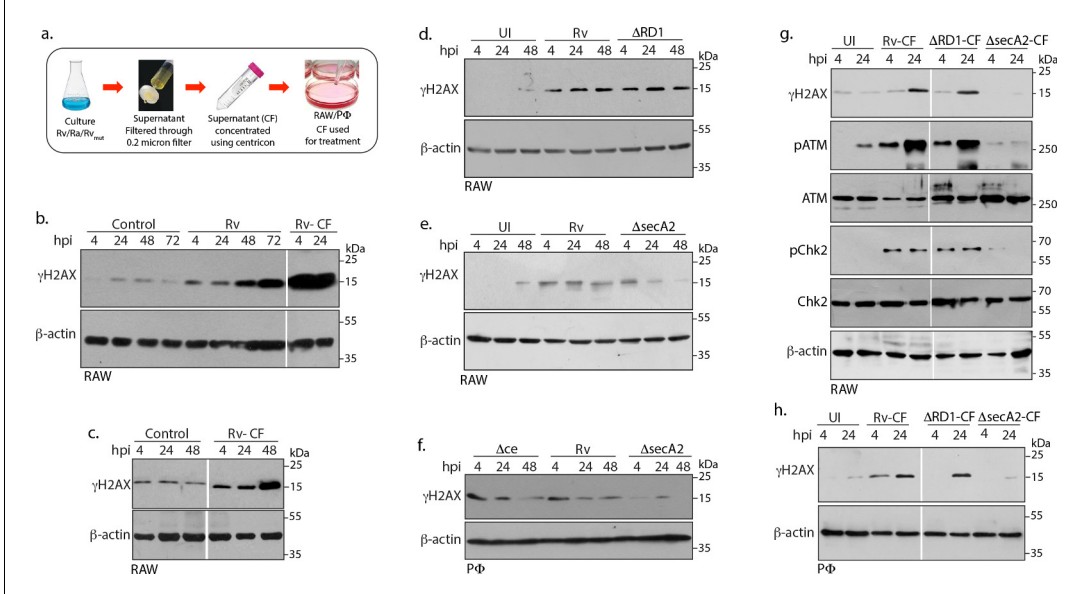

**Figure 4.** Mtb SecA2 secretome is necessary and sufficient for DNA damage. (a) Schematic depiction of culture filtrate (CF) preparation (described in Materials and methods). (b) RAW264.7 cells were either uninfected or infected with *Rv* or treated with 20 µg/ml of CF obtained from *Rv* (Rv-CF). (c) RAW264.7 cells were either left untreated or treated with 1 µg/ml of Rv-CF for 4, 24 and 48 hr. (d and e) RAW264.7 cells were infected with *Rv* or (d) *RvΔRD1* (deletion mutant of RD1 region) or (e) *RvΔsecA2* (deletion mutant of SecA2) for indicated time points. (f) PΦ cells were infected with *Rv* or *RvΔCE* or *RvΔsecA2* for indicated time points (g and h). 1 µg/ml of CF obtained from *Rv* (Rv-CF) or *RvΔRD1* (ΔRD1-CF) or *RvΔsecA2* (ΔsecA2-CF) was used to treat (g) RAW264.7 or (h) PΦ for indicated time points. (b–h) WCL from all the above conditions were subjected to immunoblotting with indicated antibodies.

survival upon ATM-I treatment is contingent on inhibition of DDR pathway. Toward this, we treated the cells with the inhibitor for immediate downstream effector Chk2. Interestingly, treatment of infected cells with Chk2 inhibitor (Chk2-I) had negligible effect on *Mtb* survival (*Figure 5d & e*; *Figure 5—source data 1*), suggesting that while ATM activation is essential; *Mtb* survival is not dependent on the activation of downstream effectors. Thus, we hypothesized that ATM activation upon infection may be channeling survival signals through alternate pathways.

## Inhibition of ATM-Akt axis hypersensitizes host cells to apoptosis

In addition to ATM and ATR, H2AX can also be phosphorylated by DNA-PK catalytic subunit (DNA-PKcs), the third PIKK that also coordinates DDR. DNA-PKcs is recruited to the DSBs at the heterodimer complex of Ku70/80 to form the holoenzyme DNA-PK, wherein it initiates the repair. Infection of PΦ with *Rv* resulted in the activation of DNA-PKcs, albeit to much lower levels and at later time points compared with the ATM (*Figure 6a*). Results in *Figure 5a* showed presence of residual γH2AX despite ATM-I treatment, thus we sought to assess the effect of combinatorial inhibition of ATM and DNA-PK or ATR on the levels γH2AX. PΦ treated with ATM-I or ATM-I + DNA-PK-I (NU7441; inhibitor of DNA-PK) or ATM-I + ATR-I(VE-281; inhibitor of ATR) were infected with *Rv* and the extent of H2AX phosphorylation was evaluated (*Figure 6b*). PΦ irradiated at 2Gy was used as the positive control (*Figure 6—figure supplement 1*). While the addition of ATR-I to ATM-I did not alter the residual levels of γH2AX, addition of DNA-PK-I to ATM-I resulted in further decrease in the levels of γH2AX, corroborating data in *Figure 6a*. Next, we assessed the role of DNA-PK and ATR in the survival of pathogen inside the host. Toward this, host cells treated with inhibitors individually were infected with *Rv* and were enumerated for CFUs (*Figure 6c* and *Figure 6—source data 1*). It is evident that addition of DNA-PK-I or ATR-I did not alter the CFUs, suggesting that even though DNA-PK seems to be activated upon *Mtb* infection, it does not influence the survival of the pathogen (*Figure 6c* and *Figure 6—source data 1*). Taken together, the data suggests that ATM might be feeding in to some parallel pathway which is necessary for the intracellular survival of *Mtb*.

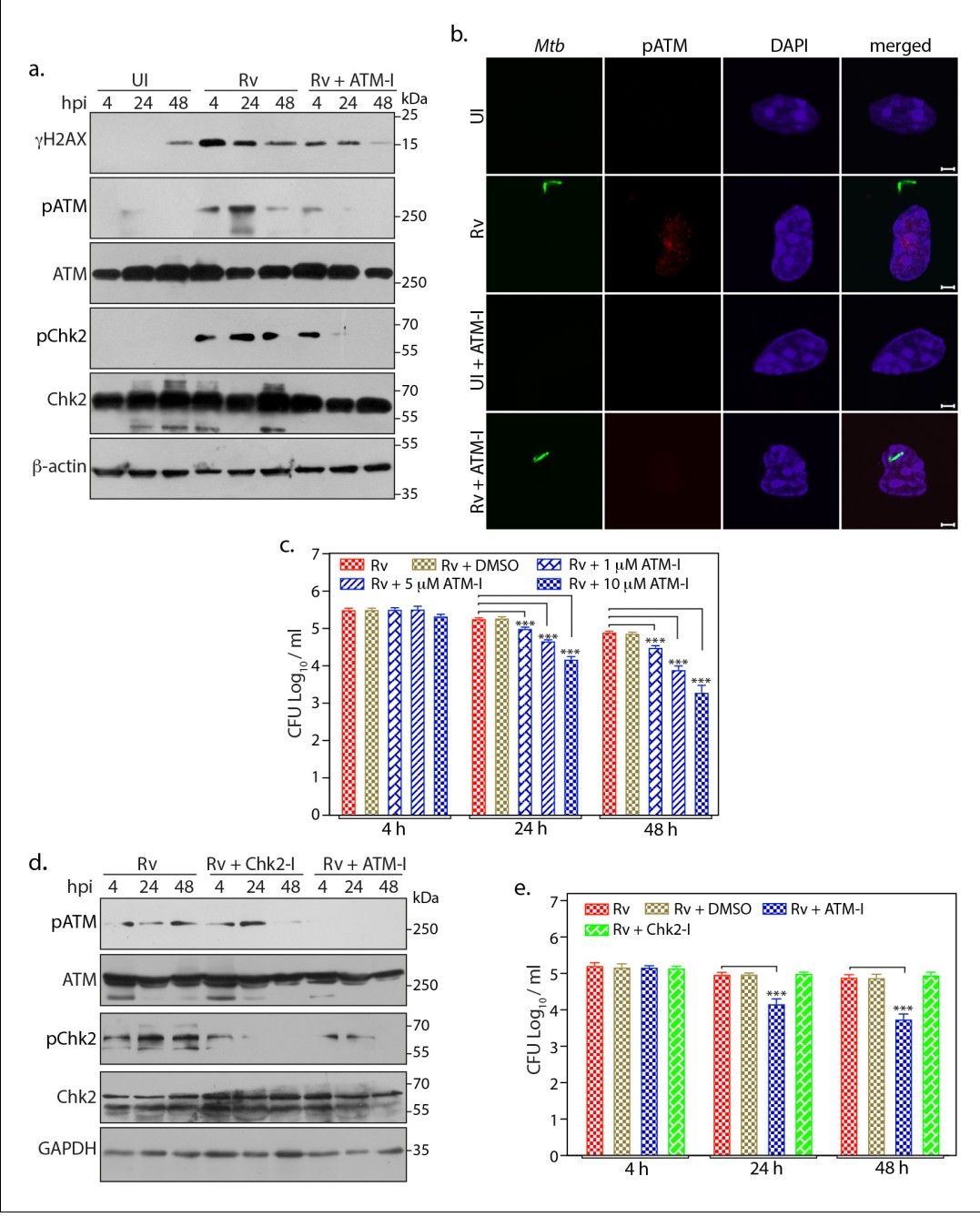

**Figure 5.** ATM activation provides survival advantages to Mtb. (**a**) PΦ were pre-treated with 10 µM of KU55933 (ATM-I) for 1 hr prior to *Rv* infection. 4 h p.i, extracellular bacilli were washed off and cells were replenished with fresh media containing ATM-I for 24 or 48 hr. Cells were lysed with RIPA to prepare WCL which were then subjected to immunoblotting against indicated antibodies. (**b**) Representative images of immunofluorescence showing pATM(S1981) foci formation (AlexaFluor594-Red) in PΦ infected with GFP-*Rv* (Green) in the presence or absence of ATM-I for 24 hr p.i. Nuclei were stained with DAPI. Images were captured at 100X magnification. Scale bar: 2 µm. (**c**) PΦ were infected with *Rv* and treated with 1, 5 or 10 µM of ATM-I as described above. Cells were lysed with 0.05% SDS to release the intracellular *Mtb* bacilli and CFUs were enumerated. (**d–e**) PΦ were pre-treated with 10 µM ATM-I or Chk1/2 inhibitor (Chk2-I) or DMSO (vehicle). Cells were lysed either with RIPA or 0.05% SDS. (**d**) RIPA WCL were subjected to immunoblotting using α-pATM, α-ATM, α-pChk2(T68), α-Chk2 and α-GAPDH antibodies. (**e**) SDS lysates were serially diluted and plated on 7H11 plates to determine CFUs.
The online version of this article includes the following source data and figure supplement(s) for figure 5:

**Source data 1.** ATM activation provides survival advantages to Mtb.

*Figure 5 continued on next page*

*Figure 5 continued*

**Figure supplement 1.** Minimum inhibitory concentration (MIC).

An established survival strategy employed by *Mtb* is to circumvent host apoptosis and upregulate pro-survival signals (*Srinivasan et al., 2014*). PI3K/Akt pathway, which upon activation inhibits apoptosis and promotes cell survival, has been shown to be up regulated during *Mtb* infection (*Figure 6d*; *Lachmandas et al., 2016*). Western blot analysis showed that treatment with either ATM-I or Akt-I inhibited the activation of ATM and Akt, respectively (*Figure 6e*). Data suggests that administration of Akt inhibitor (Akt-I) alone or in combination with ATM-I significantly reduced the intracellular survival of *Mtb* (*Figure 6f*). Treatment with a combination of ATM-I + Akt-I showed survival defects similar to Akt-I alone indicating that ATM mediated signals are channeling through Akt (*Figure 6f* and *Figure 6—source data 1*). While treatment of cells with ATM-I resulted in reduced activation of Akt, addition of Akt-I did not impact activation of ATM, suggesting that ATM activation is upstream of Akt activation (*Figure 6e*). Interestingly, the impact of Akt inhibitor on the survival of *Rv* was higher compared with ATM-I (*Figure 6f*; *Figure 6—source data 1*). Moreover, while the addition of ATM-I decreased activation of Akt, the reduction with Akt-I was higher, suggesting that other signaling pathways could also be involved in activating Akt (*Figure 6e*).

There exists a possibility that the survival differences observed could be due to reduced viability of host cells in the presence of inhibitor. To negate the cytotoxic effects of inhibitors, we evaluated the survival of uninfected and *Rv* infected PΦ in the presence of different concentrations of ATM-I or Akt-I or Chk2-I (*Figure 6—figure supplement 2*). Treatment of uninfected PΦ with ATM-I or Akt-I showed reduced viability, however, infection of PΦ with *Rv* marginally decreased the survival, compared with the UI cells, suggesting that the modulation of pathogen survival is not due to cytotoxicity (*Figure 6f* and *Figure 6—figure supplement 2*). To analyze the impact of ATM-I or Akt-I or ATM-I + Akt-I on the apoptosis, we performed flow cytometry analysis of UI and *Rv* infected cells in the absence and presence of inhibitors using Annexin V/7-AAD staining. While the addition of inhibitors did not impact the percent live cells (*Figure 6g*; top panel and *Figure 6—source data 1*) in UI, their presence decreased the percent live population in *Rv*-infected cells (*Figure 6g*; top panel and *Figure 6—source data 1*). We observed concomitant increase in the early and late apoptotic cells in the presence of ATM-I or Akt-I or ATM-I + Akt-I (*Figure 6g*; bottom panel and *Figure 6—source data 1*). Thus, the data suggests that *Rv* inflicts DSBs to activate ATM which in turn activates Akt resulting in anti-apoptotic and pro-survival signals which favors *Mtb* survival (*Figure 7g*).

## ATM kinase is a potential therapeutic target for host directed TB therapy

Persistent DNA damage drives genomic instability (*Tubbs and Nussenzweig, 2017*). Continuous DNA damage triggered during prolonged *Mtb* infection in established and primary cell lines encouraged us to examine the occurrence of possible DSBs in *Mtb* infected mice lungs. BALB/c mice challenged with *Rv* strain through aerosolic route were evaluated for bacillary load at 1- and 56 days p.i (*Figure 7a* and *Figure 7—source data 1*). We observed appearance of granulomatous lesions and significant bacillary load at 56 days p.i suggesting that *Mtb* has established chronic infection in the lungs (*Figure 7—figure supplement 1a*). Subsequently, tissue lysates prepared from uninfected and *Rv* infected mice lungs were subjected to western blotting to evaluate the expression of molecular markers corresponding to DNA damage and DDR. While the levels of γH2AX, pATM, pChk2, and pAkt was negligible in the uninfected mice lungs, we observed significantly elevated levels of γH2AX, pATM, pChk2 and pAkt in the lungs of *Rv* infected mice, suggesting that *Mtb* infection produces sustained DSBs and Akt mediated proliferation (*Figure 7b* and *Figure 7—figure supplement 1b*). In a similar vein, we also observed increased levels of γH2AX in the lysates obtained from spleen of infected mice (*Figure 7b*; bottom panels). CFUs in spleen after 56 days of infection suggests dissemination of *Rv* from lungs (*Figure 7a*). Collectively the results suggest that pathogenic *Mtb* induces genotoxicity both ex vivo and in vivo resulting in deleterious DSBs in the host genome.

DSBs activate ATM, which in turn phosphorylates H2AX to mark the damaged site and activates Chk2 and p53 to alter the cell cycle progression. In parallel, it also activates Akt to delay cell death and promote survival of the pathogen (*Figure 7b*). We hypothesized that *Mtb* mediated

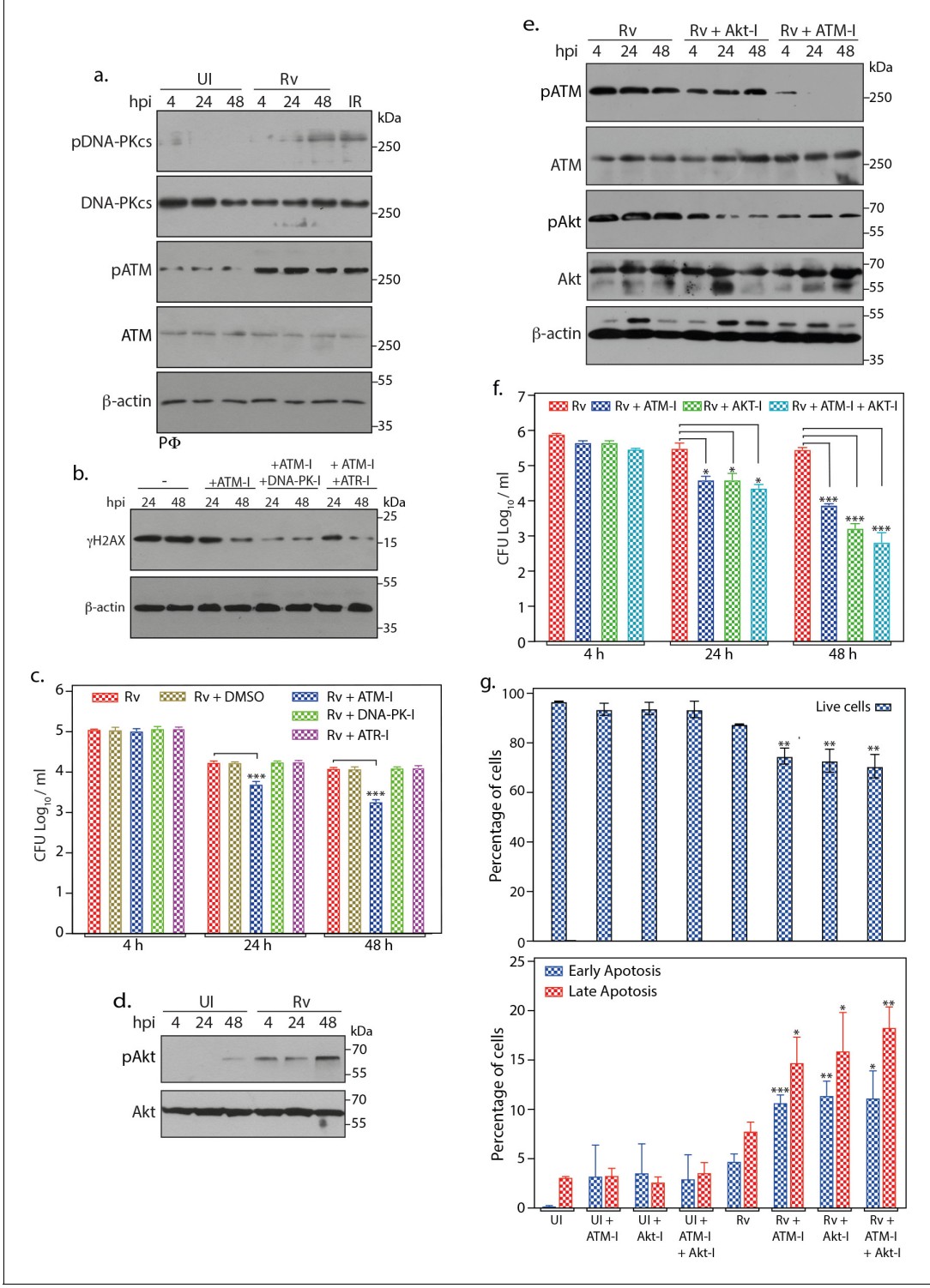

**Figure 6.** Inhibition of ATM-Akt axis hypersensitizes host cells to apoptosis. (a) To determine the activation of DNA-PKcs upon *Rv* infection, PΦ were infected with *Rv* as indicated and WCL were immunoblotted with α-pDNA-PKcs-S2056, α-DNA-PKcs, α- pATM(S1981), α-ATM, and α-β-actin. WCLs prepared from PΦ irradiated at 2Gy was used as positive control. (b) PΦ were pre-treated with 10 µM ATM-I or 10 µM ATM-I + 1 µM DNA-PK-I or 10 µM ATM-I + 10 µM ATR-I for 1 hr prior to *Rv* infection. 4 h p.i, extracellular bacilli were washed off and cells were replenished with fresh media containing above inhibitors with indicated combinations for 24 and 48 hr. Cells were lysed with RIPA to prepare WCL which were then subjected to immunoblotting against α-γH2AX and α-β-actin antibodies. (c) PΦ were infected with *Rv* and treated with DMSO or 10 µM ATM-I or 1 µM DNA-PK-I or 10 µM ATR-I individually as described above. Cells were harvested at indicated time points and CFUs were enumerated as described previously. (d)

*Figure 6 continued on next page*

*Figure 6 continued*

WCL from PΦ infected with *Rv* for indicated time points were immunoblotted with α-pAkt (S473), α-Akt antibodies (**e–f**) PΦ were pre-treated with either 10 µM of ATM-I or Akt inhibitor (Akt-I) or ATM-I + Akt-I for 1 hr prior to *Rv* infection. 4 hr p.i., extracellular bacilli were washed off and cells were replenished with fresh media containing inhibitors. At defined time points, cells were lysed either with RIPA or 0.05% SDS. (**e**) WCL were subjected to immunoblotting with α-pATM(S1981), α-ATM, α-pAkt (S473), α-Akt and α-β-actin antibodies. (**f**) Serially diluted SDS lysates were plated on 7H11 to determine CFU. (**g**) At 48 hr time point cells were scraped in PBS and subsequently stained with Annexin V-FITC and 7AAD. Cells were analyzed by flow cytometry to determine the percentage of live cells (upper panel) or cells undergoing early/late apoptosis (lower panel). Inhibitor treatments were as described above. Readings are average of three biological replicates. Error bar, SD. *, p≤0.05; **, p≤0.005; ***, p≤0.0005. Significance was calculated with respect to *Rv*.

The online version of this article includes the following source data and figure supplement(s) for figure 6:

**Source data 1.** ATM inhibition hypersensitizes host cells to apoptosis.
**Figure supplement 1.** Confirming the activity of inhibitors.
**Figure supplement 2.** Cell viability upon treatment with inhibitors.

genotoxicity provides a survival niche through activation of ATM kinase (*Figure 7b*). We sought to explore the possibility of utilizing ATM inhibitor ATM-I as a possible candidate that can be used in combination with INH towards adjunct **h**ost **d**irected **t**herapy (HDT) (*Rayasam and Balganesh, 2015*). To investigate this possibility, PΦ were infected with *Rv* and the infection was allowed to establish for 24 hr. The cells were then treated with either vehicle or INH or ATM-I or INH +ATM-I and the CFUs were enumerated at 24, 48 and 72 hr post treatment. The treatment of *Rv* infected cells with ATM-I or INH alone decreased the survival of pathogen by ~2 or 4-fold, respectively (*Figure 7c* and *Figure 7—source data 1*). Notably, treatment of *Rv* infected cells with ATM-I + INH decreased the pathogen survival by ~11 fold. This encouraged us to investigate the possibility of utilizing ATM inhibitor toward adjunct HDT for TB with the help of murine infection model. Mice were aerosolically infected with *Rv* and 15 days p.i mice were treated with vehicle or INH or ATM-I or INH +ATM-I for the next 15 days followed by CFU enumeration in both lungs and spleen (*Figure 7d*). While treatment with INH alone resulted ~1 log fold reduction in the *Mtb* load, treatment with ATM-I alone did not show any difference. Notably, combination of ATM-I + INH treatment resulted in ~1 log fold better clearance compared with INH treatment alone (*Figure 7e* and *Figure 7—source data 1*). The impact of ATM-I+INH treatment compared with INH treatment alone was more evident in the spleen (*Figure 7f* and *Figure 7—source data 1*), suggesting combined therapy might have compromised dissemination of the pathogen. Taken together, we propose that ATM inhibitor is a potential candidate for HDT.

## Discussion

The present study was designed to address the following questions: i) Does *Mtb* target the genomic integrity of the host? If so, do virulent and avirulent *Mtb* target the genomic integrity of the host differentially, and in that case, how? ii) How the host cell responds to the DNA damage mediated through *Mtb*? iii). What is the role of the *Mtb* secretome, if any, in imparting DNA damage? iv). What is the role of ATM kinase activation and subsequent downstream signaling in the survival of pathogen within the host? vi). Is this *Mtb* mediated DNA damage manifested in the mice model of in vivo infection?

Results show that *Mtb* infection promotes γH2AX upregulation in ex vivo models of murine/human origin and murine primary macrophages. The induction of γH2AX and subsequent formation of foci could also be detected by immunofluorescence (*Figure 1a–h*). These results are in accordance with earlier studies wherein induction of γH2AX foci and genomic instability was detected through immunofluorescence and in situ hybridization, respectively, in the ex vivo model of *Mtb* infection (*Castro-Garza et al., 2018*; *Mohanty et al., 2016*). While the induction of γH2AX is dependent on the presence of the live bacteria, *Ra* a non-pathogenic counterpart of *Rv* could also induce γH2AX levels, albeit at comparatively slower rates (*Figure 1d–g*). In addition to temporal differences, *Ra* and *Rv* appear to inflict distinct types of damages resulting in the activation of ATR and ATM pathways, respectively (*Figure 2b*). The infection with *Rv* results in lingering γH2AX levels even at later time points, suggesting the sustained presence of pathogenic *Mtb* continues to inflict DNA damage (*Figure 3a*). In consonance with this hypothesis, elimination of intracellular *Mtb* drastically

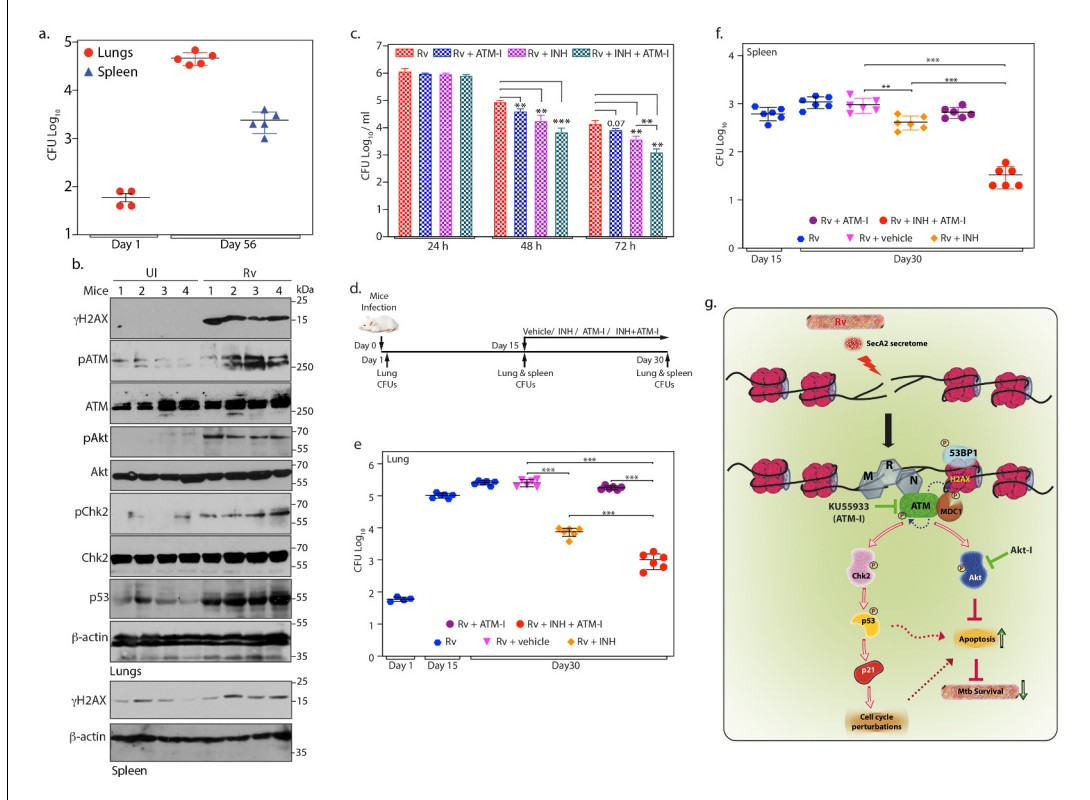

**Figure 7.** ATM kinase is a potential therapeutic target for host directed TB therapy. (**a**) Graphical representation depicting *Rv* load at day 1 (n = 4) and day 56 (n = 5) of infection in lungs and spleen of mice. Lung homogenates prepared at indicated time points were plated on 7H11 plates to enumerate CFU per lung or per spleen (**b**) Lung and spleen lysates were prepared from uninfected and *Rv* infected mice 56 days p.i. 100 μg lung lysates were subjected to immunoblotting with α-γH2AX, α-pATM(S1981) α-ATM, α-pCHK2(T68), α-CHK2, α-pAkt (S473), α-Akt, α-p53 and α-β-actin antibodies. (**c**) PΦ were infected with *Rv* at high MOI (1:10) as described above. 24 h p.i, cells were treated with either 3.6 μM isoniazid (INH) or 10 μM ATM-I alone or INH + ATM-I together. CFUs were enumerated at 24, 48 and 72 hr post treatment. (**d**) Schematic representation of the mice infection and drug treatment protocol used. (**e–f**) CFUs were enumerated in lungs of mice at Day 1, and in the lungs (**e**) and spleen (**f**) on day 15 and 30 of infection. Number of mice in each batch was 6 (n = 6) except in *Rv* at Day 1(n = 5). Error bar, SD. *, p≤0.05; **, p≤0.005; ***, p≤0.0005. (**g**) Model depicting the findings. *Rv* induces genotoxicity and causes deleterious DSBs in the host genome through SecA2 secretome. Host cell in response to the occurrence of DSBs activate ATM kinase and is recruited at the site of damage by the sensor, MRN complex. Activated ATM autophosphorylates itself and phosphorylates H2AX in the chromatin flanking the sites of DNA damage, which becomes the foundation for the recruitment of mediator protein MDC1, thus amplifying DDR. pATM promote recruitment of 53BP1 at the damage site. pATM as a part of DDR also activates downstream effectors, Chk2 and p53, which are responsible for alterations in the host cell cycle. pATM in a parallel pathway also activates Akt, which is known inhibit apoptosis and promote cell survival. Activation of ATM and Akt and subsequent inhibition of apoptosis provides survival advantage to *Rv*. Inhibition of ATM or Akt activation through inhibitors, ATM-I or Akt-I, respectively, promote host cell apoptosis, which impedes the bacilli growth. Phosphorylation status of proteins is depicted with a P in a circle.

The online version of this article includes the following source data and figure supplement(s) for figure 7:

**Source data 1.** ATM kinase is a potential therapeutic target for host directed TB therapy.
**Figure supplement 1.** Gross pathology and fold change in the expression of host proteins.
**Figure supplement 2.** Graphical representation showing the fold change expression of the indicated protein.

reduced γH2AX levels/DNA damage, suggesting *Mtb* may be inducing DNA damage at a faster frequency relative to the host's ability to repair them (*Figure 3c*).

In response to the DSBs generated upon pathogenic *E. coli* infection, host activates phosphorylation of Cdc25 through ATM-Chk2 pathway to hamper cell cycle progression. However, at higher MOI, host cell undergoes p53 SUMOylation through SENP1 downregulation to promote senescence (*Cougnoux et al., 2014*). On the other hand, *N. gonorrhoeae* decelerate the host cell cycle by reducing p53 and concurrent elevation of p21 and p27 (*Vielfort et al., 2013*). *L. monocytogenes* in addition to impairing host MRE11 also activates DDR in ATM/ATR independent fashion eventually resulting in cell cycle delay (*Leitão et al., 2014*). Taken together, it is evident that pathogens utilize

diverse and distinct mechanisms to perturb the host cell cycle to avoid untimely death. We observed activation of ATM-Chk2 pathway (*Figure 2b*) and subsequent decrease in the cells in G1 phase (*Figure 3e*), an indicator of differed host cell cycle at the early stages of *Mtb* infection. However, at the later time points, infected cells showed increased cell death, most likely an effort by the host to curb the transmission of genotoxic effects (*Figure 3e*). Interestingly, cell cycle perturbations were limited to the cells possessing intracellular bacilli but not extended to bystander-uninfected cells (*Figure 3e*). Thus, our results demonstrate that *Mtb* mediated DNA damage disturbs the homeostatic coordination among the cellular components participating in cell cycle.

Toxins like Cytolethal distending toxin (CDT), typhoid toxin and colibactin from pathogenic bacteria are examples of characterized toxins that are reported to block the host cell cycle progression after inducing DNA damage in the host (*El-Aouar Filho et al., 2017*). Persistent presence of DSBs and modulation of host cell cycle in cells harboring bacilli, propelled us to examine the role of *Mtb* secretory factors in imparting genotoxicity. The proteins exported from *Mtb* are considered a source of virulence, however not many secreted proteins have been individually characterized. *Mtb* secretes >300 secretory factors with the help of secretory systems such as classical TAT and SecA1; accessory SecA2 and TypeVII. Results presented in *Figure 4e–h* suggest that SecA2 mediated secretory factors are principally responsible for the DSBs in the host. Mass spectrometry-based study has identified the secretion of 37 proteins, including Mce1, Mce4 family and multiple solute binding lipoproteins to be dependent on the SecA2 pathway. In addition to these, secretion of protein kinase G (PknG), superoxide dismutase (SodA) and phosphatase SapM proteins are also through SecA2 pathway (*Feltcher et al., 2015*; *Miller et al., 2017*; *Sullivan et al., 2012*; *Zulauf et al., 2018*). As polyketides and lipids from mycobacteria were suggested to be cyclomodulins; therefore, we speculate that one or more lipoproteins secreted through the SecA2 pathway could be critical for genotoxicity (*Cumming et al., 2017*; *George et al., 1998*). Macrophages generate reactive oxidation species as an anti-microbial activity to combat the survival of *Mtb* intracellularly. Mice lacking p47 and *gp91* subunit of NADPH oxidase (gp91$^{phox-/-}$) are susceptible to *Mtb* as the macrophages in these mice fail to generate ROS (*Cooper et al., 2000*). While the primary purpose of generated ROS is to clear the intracellular bacilli, it can however damage its own DNA as exemplified in case of *C. trachomatis* infection (*Chumduri et al., 2013*; *Yu and Anderson, 1997*). On the contrary, *H. pylori*, *L. monocytogenes* mediated DSBs are ROS-independent. SodA$_{Mtb}$, which is secreted through SecA2 pathway plays an important role in detoxification of ROS generated inside the host thus partially neutralizing the anti-microbial property of ROS (*Chan et al., 1992*; *Lau et al., 1998*; *Piddington et al., 2001*). In *RvΔSecA2* mutant SodA$_{Mtb}$ secretion is expected to be compromised and hence one would expect higher levels of host ROS upon infection. The fact that *RvΔSecA2* mutant fails to inflict DSBs suggest that *Mtb* mediated damage is unlikely to be ROS-dependent. In agreement with this, addition of N-acetyl cysteine (NAC), a scavenger of ROS, to the infected cells did not alter the γH2AX, suggesting that DNA damage is not mediated through ROS (*Figure 7—figure supplement 1c*). Our future study would be aimed towards identifying the specific effector(s) and the mechanism of how the effector brings about DSBs in the host.

*Mtb* is known to modulate multiple host signaling pathways to establish favorable environment in the host for its survival. Moreover, a previous study (*Hinchey et al., 2007*) showed that *RvΔsecA2* strongly induced markers of apoptosis in both human and murine macrophages suggesting a critical role played by SecA2 secretome in preventing apoptosis. Thus, we investigated the role of *Mtb* induced host DNA damage and subsequent activation of ATM-Chk2 pathway in facilitating and enhancing its survival. While presence of ATM inhibitor significantly compromised the *Mtb* survival, addition of inhibitor for one of the downstream effectors, Chk2 which is associated to cell cycle checkpoints activation, failed to do so (*Figure 5e*). These results suggest that while the survival signals for *Mtb* are dependent on activation of ATM, it is not channeled through the classical DDR. High-throughput study revealed that following DSBs, ATM-dependent phosphorylation of ~700 substrates regulate multitude of cellular processes such as cell cycle, cellular differentiation etc., emphasizing its global regulatory role. Importantly, DDR was suggested to crosstalk with the IGF-1/PI3K/Akt pathway to elicit cell survival signals, which is also supported by data that demonstrated ATM-dependent phosphorylation of Akt at S473 (*Khalil et al., 2011*; *Matsuoka et al., 2007*). Previous reports suggested that inhibiting Akt diminished *Mtb* survival inside host cells and proposed it to be a novel target for host directed therapy (*Kuijl et al., 2007*). Thus, we hypothesized a probable divergence of ATM signaling towards Akt activation may play a role in influencing *Mtb* survival. We

observed that addition of Akt-I compromises the survival of *Mtb* (*Figure 6f*). Furthermore, combinatorial addition of both ATM and Akt inhibitors did not show either additive or synergistic impact on the *Mtb* survival, suggesting that survival signals are routed through ATM-Akt axis (*Figure 6f*).

*Mtb* infection of the host cells can induce host cell death through both apoptotic and non-apoptotic pathways. Apoptotic death is considered to be advantageous for the host as it results in the elimination of pathogen (*Fratazzi et al., 1999*; *Gan et al., 2008*; *Molloy et al., 1994*). Induction of non-apoptotic death of the host cells is beneficial for the bacilli as it fails to eliminate the pathogen (*Dobos et al., 2000*; *Keane et al., 2000*; *Lee and Paull, 2005*; *Martin et al., 2012*; *Park et al., 2006*). Recently, it was reported that internalization of *Mtb* aggregates by the host cell results in killing of the host and there was a direct correlation between the size of aggregates and cell death. However, the death in such cases was through non-apoptotic pathway (*Mahamed et al., 2017*). In our experiments we kept MOI at one to circumvent aggregation and non-apoptotic cell death of the host. We observed significantly increased apoptosis upon the addition of inhibitors for either ATM or Akt or both (*Figure 6g*). These results are in line with the previous findings, wherein bacterial pathogens such as *Salmonella typhimurium* and *Coxiella burnetii* have been shown to get survival advantage through Akt activation (*Kuijl et al., 2007*; *Voth and Heinzen, 2009*).

We hypothesize that persistent DSBs caused by *Mtb* could be a means to constitutively activate ATM, which in turn would activate Akt and inhibit apoptosis thus facilitating the survival of the pathogen. We corroborated our finding in the murine model of chronic tuberculosis infection, wherein the infected lungs showed the activation of γH2AX and ATM-Chk2 signaling (*Figure 7a–b*). Addition of ATM inhibitor to the INH regime substantially reduced the bacilli load in the host establishing ATM as a potent target for HDT (*Figure 7c–f*). In conclusion, our study demonstrates that *Mtb* through SecA2 secretome challenges the host fitness by damaging the genome (*Figure 7g*). In an effort to repair the damage, the host cell activates the ATM kinase mediated signaling cascade, which is exploited by *Mtb* to survive. We believe we have identified a novel survival mechanism utilized by *Mtb*, wherein the pathogen constantly challenges the host genome leading to the activation of pro-survival ATM-Akt signals (*Figure 7g*). Hence, we propose the use of ATM inhibitors as adjunct for HDT in the treatment of tuberculosis.

## Materials and methods

### Materials

Fine chemicals used in the study were purchased from Sigma-Merck. Bacterial media components were purchased from Difco-BD biosciences. All cell culture reagents were procured from Thermo fisher scientific Inc or Hyclone. α-γH2AX (2577S), α-ATM (2873S), α-pATR-S428 (2853S), α-pChk1-S345 (2348S), α-pChk2-T68 (2661S), α-Chk2 (2662S), α-CHK1(2360), α-ATR (13934), α-pAkt-S473 (9271S), α-Akt (9272S), α-p53BP1-T543 (3428S), α-MRE11(4895S), α-pNbs1-S343 and α-Nbs1 (3002S) antibodies were procured from Cell Signalling Technologies. α-pATM-S1891(ab81292) α−53BP1 (ab36823), α-pMDC1-T4 (ab35967), α-RPA2 (ab2175) and α-p53 (ab28) antibody was procured from Abcam. α-p21 (sc-817), α-β-actin (sc-47778), α-β-tubulin (sc-55529) and α-GAPDH (sc-25778) antibodies were purchase from Santacruz Biotechnology. α-DNA-PKcs (SAB4502385), α-pDNA-PKcs-S2056 (SAB4504169) were purchased from Sigma. α-MDC1 and α-pRPA2-S4/S8 was purchased from Novus Biologicals and HRP conjugated secondary antibodies were purchased from Jackson laboratories. α-MDC1 was purchased from Novus Biologicals. AlaxaFluor594 and Alexa-Fuor647 fluorophore conjugated secondary antibodies were procured from Thermo fisher scientific Inc.

### Mtb strains, cell culture and Mtb infection

*Mtb* strains, *H37Rv* (*Rv*) and *H37Ra* (*Ra*), *RvΔRD1* and *RvΔSecA2* were propagated in Middlebrook 7H9 broth (Difco) supplemented with 10% ADC (Albumin-Dextrose-Catalase complex), 0.2% glycerol (v/v), 0.05% (v/v) Tween-80 with shaking at 100 rpm at 37˚C. *Rv* and *Ra* strains were electroporated with pSC-301-GFP plasmid (*Cowley and Av-Gay, 2001*) to generate GFP expressing strains. Human acute monocytic leukemia cell line, THP-1 (ATCC) and RAW264.7 (ATCC), peritoneal macrophages (PΦ) derived from Balb/c mice were grown in the cell culture media (RPMI-1640 constituted with 1X Anti-Anti and 10% heat inactivated (HI)-fetal bovine serum) at 37˚C. THP-1 cells were treated with 10

ng/mL PMA for 24 hr followed by 12 hr resting without PMA before *Mtb* infection. RAW264.7 cells were seeded 12–15 hr before *Mtb* infection. PΦ were extracted from thioglycolate injected Balb/c mice as described earlier (*Zhang et al., 2008*). PΦ isolated from the peritoneal cavity were resuspended in cell culture media and infected 12–15 hr post plating with *Mtb*. THP1 cells used in the study have been authenticated by STR profiling. All the cell lines used were free of mycoplasma contamination. Single-cell suspensions of actively growing *Mtb* strains were prepared and all infections were performed at MOI of 1, else indicated. Four hours p.i extracellular bacilli were removed by washing thrice with PBS and the cells were supplemented with fresh cell culture media for defined periods of time. For the inhibitor studies, RAW or PΦ were pre-treated with different doses (1, 5 or 10 µM) of ATM kinase inhibitor (KU55933, SML1109, Sigma; S1092, Selleckchem) or Chk2 inhibitor (C3742, Sigma) or Akt1/2 kinase inhibitor (A6730, Sigma), ATR inhibitor (VE-281, SML1415, Sigma), DNA-PK inhibitor (NU7441, 18003649897, Cayman chemical company). ATM-I, DNA-PK-I and ATR-I were used at reported dose of 10 µM, 1 µM and 10 µM, respectively. 4 hr p.i cells were supplemented with fresh media containing inhibitors. For the clearance of intracellular *Mtb* from infected RAW cells, cells were treated with 1 µg/ml (7.3 µM) of isoniazid (INH). Colony forming units (CFUs) were enumerated at 24, 48 and 72 hr p.i to evaluate the bacterial load. RAW/PΦ were lysed in 0.05% SDS and different dilutions of the lysates were plated on 7H11 agar plates supplemented with 10% OADC (oleic acid- Albumin-Dextrose-Catalase complex). After 21 days, CFUs of *Mtb* were enumerated.

## Preparation of lysates, western blotting, and immunofluorescence

Cell lines were lysed and snap-frozen mice lungs and spleen were macerated in cold RIPA buffer (50 mM Tris, pH 8.0, 150 mM NaCl, 1.0% NP-40, 0.5% Sodium deoxycholate, 0.1% SDS, freshly supplemented with PhosSTOP and Complete protease inhibitor (Roche). The lysates were resolved in 10–15% SDS-PAGE followed by transfer to nitrocellulose membrane (Biorad). Antibody dilutions, incubation time and temperature conditions were according to the manufacturers' protocol. Blots were developed on autoradiograms using chemiluminescent HRP substrate (ECL, Millipore). For immunofluorescence assay, $5 \times 10^5$ cells were seeded on sterile coverslips in 6-well tissue culture plates and infected with GFP expressing *Rv/Ra*. At defined time points, cells were fixed with 2% PFA and immunostaining was performed according to the manufacturers' protocol provided by respective antibody. Coverslips were mounted on slides with DAPI containing Vectashield mountant (H-1200, Vector Labs). Images were captured in Carl Zeiss LSM510 Meta confocal microscope.

## Animal infection

Actively growing *Rv* cultures were used for aerosolically infecting mice to implant 200 CFU/lung of Balb/c mice (6–8 weeks old of either sex) inside Madison Aerosol Chamber (University of Wisconsin, Madison, WI). Infection load in lungs and spleen was determined at day 1 and 56 p.i to evaluate deposition and commencement of chronic infection of *Mtb*, respectively. One lung and part of spleen of infected mice were homogenized and plated on 7H11 plates supplemented with PANTA (BD Bioscience) to enumerate CFUs. Second lung and part of the spleen from infected and uninfected mice were used preparation of tissue lysates. In another experimental set up, Balb/c mice were infected with *Rv* as stated above. Post 15 days of infection, mice were either kept untreated or treated with Vehicle (5% DMSO in sterile PBS), INH (prepared in sterile water, 25 mg/kg/dose through oral gavage), KU55933 (prepared in DMSO and further diluted in PBS such that DMSO is 5% (v/v) in PBS before injecting in the peritoneal cavity; administered every third day at 10 mg/Kg/dose [*Batey et al., 2013*]) or INH+ATM-I according to the dosage stated for next 15 days. Five batches of six mice each (No treatment, Vehicle, INH, ATM-I or INH+ATM-I) at 30 day p.i were used for assessing *Mtb* CFUs in lungs and spleen.

## Cell cycle analysis and apoptosis assay

RAW cells infected with GFP expressing *Rv* at MOI of 1 for indicated time points. Cells were fixed with 70% ethanol for overnight at 4°C followed by PBS washes and staining with 10 µg/ml propidium iodide (PI) solution (PBS+0.05% Triton-X + 100 µg/ml RNase A). Cells were analyzed through flow cytometry (FACS Calibur, Becton Dickinson, USA) to assess the distribution of cells in different phases of the cell cycle. For the apoptosis assay, PΦ infected with *Rv* and treated with the indicated

inhibitors were stained with FITC Annexin V-7AAD according to manufacturer's protocol (BioLegend). Following staining, cells were fixed with 2% PFA before assessing the cell percentage undergoing apoptosis through flow cytometry.

## Culture filtrate protein (CFP) preparation

*Mtb* strains were cultured in modified sauton's medium (2.9 mM $KH_2PO_4$, 4.2 mM $MgSO_4$, 10.4 mM citric acid, 0.2 mM Ferric ammonium citrate, 6% Glycerol, 30.5 mM asparagine, pH 7.2) till mid-log phase. Culture supernatants were passed through 0.22 μM filter and concentrated to 1/100th of the original volume using Amicon concentrators with a 3 kDa cut off (*Rosenkrands and Andersen, 2001*). The proteins concentration in the obtained culture filtrates was quantitated.

## Statistical analysis

GraphPad Prism eight software and MS office Excel 2010 was used to execute statistical calculations. Unpaired *student's t test* was used to determine the *p-values*. Significant p values of the data sets were considered significant *$p<0.05$, **$p<0.005$, ***$p<0.0005$ and not significant (ns) if $p>0.05$. Densitometric analysis of important western blots (*Figure 7—figure supplement 2*) was performed using Image J software (Schneider et al.).

# Acknowledgements

This work was supported by the funding provided by Department of Biotechnology, Government of India (BT/PR13522/COE/34/27/2015) and National Institute of Immunology (NII) core funding. SL is thankful to NII Young Investigator fellowship and UGC-DS Kothari postdoctoral fellowship. SS acknowledges National Institute of Immunology Core funding and JC Bose Fellowship (JCB/2018/000013) for financial assistance. We thank the bio-containment facility (BSL3), central FACS and confocal facilities at NII. We thank Dr. Miriam Braunstein and Dr. Dheeraj Kumar for their kind gifts of *RvΔsecA2* and *RvΔRD1* strains of *Mtb*, respectively. Mice infection experiments were carried out at DBT funded Tuberculosis Aerosol Challenge Facility at ICGEB, Delhi, India. We thank Dr. Lakhyaveer Singh and his team for their help during animal infection experiments at TACF, ICGEB. We thank Dr. Swati Saha for her critical reading and helpful suggestions.

# Additional information

### Funding

| Funder | Grant reference number | Author |
|---|---|---|
| Department of Biotechnology, Ministry of Science and Technology | BT/PR13522/COE/34/27/2015 | Vinay Kumar Nandicoori |
| University Grants Commission | DS Kothari Postdoctoral Fellowship | Savita Lochab |
| Department of Science and Technology, Ministry of Science and Technology | JCB/2018/000013 | Sagar Sengupta |
| National Institute of Immunology | | Sagar Sengupta |

The funders had no role in study design, data collection and interpretation, or the decision to submit the work for publication.

### Author contributions

Savita Lochab, Conceptualization, Formal analysis, Investigation, Visualization, Methodology, Writing - original draft, Writing - review and editing; Yogendra Singh, Funding acquisition; Sagar Sengupta, Conceptualization, Formal analysis, Supervision; Vinay Kumar Nandicoori, Conceptualization, Formal analysis, Supervision, Funding acquisition, Project administration, Writing - review and editing

## Author ORCIDs
Savita Lochab  https://orcid.org/0000-0002-8153-7867
Yogendra Singh  https://orcid.org/0000-0002-3902-4355
Sagar Sengupta  https://orcid.org/0000-0002-6365-1770
Vinay Kumar Nandicoori  https://orcid.org/0000-0002-5682-4178

## Ethics
Animal experimentation: The experimental protocol for the animal experiments was approved by the Animal Ethics Committee of the National Institute of Immunology, New Delhi, India. The approval (IAEC#409/16 & IAEC#462/18) is as per the guidelines issued by Committee for the Purpose of Control and Supervision of Experiments on Animals (CPCSEA), Government of India.

## Decision letter and Author response
Decision letter https://doi.org/10.7554/eLife.51466.sa1
Author response https://doi.org/10.7554/eLife.51466.sa2

## Additional files
### Supplementary files
- Transparent reporting form

## Data availability
Numerical data for graphs is provided in the source data files.

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

## Appendix 1

# Supplementary methods

## Cell viability assays

MTT Assay was performed in triplicates to evaluate the cell viability during the treatment of ATM-I, AKT-I and Chk-I. PΦ were pre-treated for 1 hr with different doses (1, 5, 10 or 20 μM) ATM-I or Chk-I or Akt-I prior to infection. Post-infection cells were washed with PBS, and supplemented with fresh media with or without inhibitors. 4, 24, 48 hr p.i soluble MTT substrate (3-(4,5- dimethylthiazol-2-y1)−2,5-diphenyl tetrazolium bromide) was added at a final concentration of 500 μg/ml and cells were incubated for 4 hr. 500 μl of DMSO was added to each well, mixed and the absorbance was measured at 595 nm. Cell viability of the untreated samples were considered 100%. Cell viability in the presence of the inhibitor/Cell viability in the presence DMSO for the samples were calculated with respect to the corresponding value for the untreated sample.

Average O.D. of sample = [Average O.D. of the sample- Average O.D. of blank]

Cell viability (%) = [average O.D. of treated sample/average O.D. of untreated sample] x 100

## Minimum inhibitory concentration (MIC) determination

A stock of 73 mM of INHwas prepared in sterile water, which was serially diluted in 7H9 media to a final concentration of 73 μM. 10 mM stock of ATM-I in DMSO was used to make final concentration as 100 μM in complete 7H9 media. Actively growing *Rv* cultures at 0.6 OD were centrifuged and resuspended in fresh 7H9 complete media. 2-fold serial dilutions of INH and ATM-I in triplicates were prepared in sterile 96-well plates. The resuspended bacilli were then added in the 96-well plate such that each well receives ~10,000 bacilli. Wells with *Rv* alone was used as the control. The plates were sealed with parafilm and incubated at 37°C for 6 days followed by the addition of 20 μl resazurin (0.02% in sterile water). The plates were further incubated for 12–24 hr at 37°C till the development of color. While blue color of resazurin converts into pink for viable cells, the color remains blue in case of dead cells. MIC is defined as the lowest drug concentration that inhibits the growth bacilli. Results were assessed on the basis of color development.

## Densitometric analysis

Densitometric analysis of major findings was performed with the help of Image J software. Briefly, scans of autoradiograms (300 dpi) were opened in Image J software (*Schneider et al., 2012*) and the intensities of bands in each lane was individually measured. The intensities of γH2AX/pATM/ pATR/pChk2/pChk1/pAkt/p53 in an image were normalized with respect to their corresponding loading control β-actin /β-tubulin/GAPDH. Once the values were obtained for all the lanes in an image, the fold change was calculated as follows:

Fold Change = [Normalized intensity of a band/highest normalized intensity in that image]

