## [Decision Letter]

**Acceptance summary:**

This study provides novel insight suggesting that pathogenic mycobacteria, such as *Mycobacterium tuberculosis*, secrete a genotoxic agent through the SecA2 pathway with important implications for immune responses and pathogenesis. As such, inhibitors of the DNA damage response, include the ATP kinase described in your work, and DNA-PK inhibitors, could be used as host-directed therapy in combination with current antimicrobials to improve tuberculosis treatment outcomes. This work provides promising new avenues for further research and opens the possibility of testing existing agents, with good safety and toxicity profiles, in clinical trials as adjunctive therapy to standard TB treatment.

**Decision letter after peer review:**

Thank you for submitting your article "*Mycobacterium tuberculosis* exploits host ATM kinase for survival advantage through SecA2 secretome" for consideration by *eLife*. Your article has been reviewed by two peer reviewers, and the evaluation has been overseen by a Reviewing Editor and Wendy Garrett as the Senior Editor. The following individual involved in review of your submission has agreed to reveal their identity: Alasdair Lesley (Reviewer #1).

The reviewers have discussed the reviews with one another and the Reviewing Editor has drafted this decision to help you prepare a revised submission.

Summary:

The submission by Nandicoori explores activation of the ATM kinase, which occurs during colonization of macrophages by *Mycobacterium tuberculosis*. The authors study this in human and mouse macrophage cell lines and in primary mouse macrophages. The activation of ATM kinase led to the phosphorylation of cognate substrates Chk2 and γH2AX, the latter being detected in the form of nuclear foci, an observation that supports the induction of DNA double-strand breaks in response to *M. tuberculosis* infection. In contrast, a non-virulent strain (*H37Ra*) led to activation of the human ATR kinase, detected by its phosphorylation and phosphorylation of its substrate, Chk1. DNA damage induction could be recapitulated using bacterial supernatants and was abolished when using heat-killed bacteria or a mutant of the SecA2 secretion system. These data support the SecA2-dependent secretion of a genotoxic agent/s by virulent mycobacteria. Inhibition of ATM led to a small reduction in bacterial load in infected mouse macrophages and when combined with the antibiotic isoniazid, a synergistic effect was demonstrated. Inhibition of ATM led to apoptosis in a large proportion of infected macrophages, supporting a model where ATM promotes *M. tuberculosis* survival in its host. Inhibition of the Chk2 kinase, acting classically downstream of ATM to block cell cycle progression, did not phenocopy ATM inhibition while inhibition of Akt did. Akt has been reported as being activated by ATM but also by DNA-PK (another kinase activated together with ATM upon DNA double-strand breaks). Consistent with this, the authors report here an ATM dependent activation of Akt upon infection. These findings are of potential interest as they support the hypothesis that virulent mycobacteria secrete a genotoxic agent, that still awaits characterization, through the SecA2 pathway and suggest that inhibitors of the human DNA damage response, including ATM inhibitors and probably DNA-PK inhibitors, could be used as host-directed therapy in combination with current antimicrobials. Several of these inhibitors have already undergone clinical trials to be used alone or in combination with DNA damaging agents to cure various human cancers and some of them have demonstrated safety and efficacy.

Essential revisions:

1) In Figure 2D, phosphoATM seems nucleolar on the micrographs. This is unexpected. This could be an important finding if confirmed. To confirm signal specificity, an immunofluorescence (IF) experiment would be needed using the ATM inhibitor KU55933 to show that the nucleolar staining disappears after treatment. To confirm the localization, co-staining of phosphoATM upon infection should be performed with a nucleolar marker (e.g. nucleolin).

2) Considering that the formation of γH2AX foci is the strongest evidence for DNA double-strand break induction presented in the manuscript, pictures acquired at higher resolution showing the focal pattern of the staining should be provided. In addition, IF should also be performed using an antibody against another protein forming foci in response to DNA double-strand breaks, for example 53BP1.

3) When considering (i) the major role of NHEJ in repairing DNA double-strand break in human cells (Ciccia and Elledge, 2010) and (ii) the role of the NHEJ complex DNA-PK in activating Akt (Lan et al. Oncotarget 2016; Stronach et al. Neoplasia 2011), the impact of the inhibition of DNA-PK kinase activity on *M. tuberculosis* infection should be investigated in this study in parallel to ATM inhibition. This could be tested using one of the specific DNA-PK inhibitors available, such as NU7441.

4) After treatment with classical DNA double-strand break inducing agent, production of γH2AX is mainly ATM dependent, but has also been reported in some circumstance to be DNA-PK and/or ATR dependent. In Figure 5B, treatment with the ATM inhibitor only reduces γH2AX production, which suggests that other kinases might be at work. The authors should test the implication of ATR and DNA-PK in this residual phosphorylation, since specific inhibitors against these kinases are commercially available.

5) The authors report ATR activation upon infection by the *H37Ra* attenuated strain. IF should be performed (after *H37Ra* and *H37Rv* infection) to monitor for the accumulation of the protein RPA (complex of RPA1, RPA2 and RPA3) in the form of foci, which would support the accumulation of single-strand DNA upon infection (RPA is recruited on single-strand DNA regions).

6) Previous work has implicated ROS in the production of DNA damage in infected host cells, the role of ROS should be investigated here. An experiment with N-acetyl-cysteine or membrane permeable reduced glutathione should be performed.

7) Figure 6A and B do not provide sufficient data to support conclusions. Rather, the authors should test the effect of the KU inhibitor in vivo.

8) Furthermore, the model suggests AKT signalling promotes survival by blocking *M. tuberculosis* induced apoptosis. Therefore, *M. tuberculosis* infection with AKT or ATM inhibition should lead to increased cell death by apoptosis. However, this is very hard to assess with the data presented as, in Figure 5H, the effect of inhibitor alone is not tested (so the increased apoptosis could just be due to the inhibitor – which, as from Figure 5—figure supplement 1, does cause cell death); and in Figure 5—figure supplement 1, there is no increased cell death when *M. tuberculosis* is added. The authors should assess the effect of the inhibitors (KU and AKT-I) alone on apoptosis without *M. tuberculosis* infection.

9) Western blot data should be quantified and statistically analyzed. Some blots are sliced/edited others not in the same panel (Figure 3C); a single membrane should be used.

10) The model that decreased apoptosis is associated with increased bacterial growth is also contrary to recently published live imaging studies suggesting that macrophage death is essential for rapid bacterial growth (Mahamed et al., 2017). The authors should discuss this.

---

## [Author Response]

Essential revisions:1) In Figure 2D, phosphoATM seems nucleolar on the micrographs. This is unexpected. This could be an important finding if confirmed. To confirm signal specificity, an immunofluorescence (IF) experiment would be needed using the ATM inhibitor KU55933 to show that the nucleolar staining disappears after treatment. To confirm the localization, co-staining of phosphoATM upon infection should be performed with a nucleolar marker (e.g. nucleolin).

We concur with the observation regarding localization of pATM. In order to address the question, we performed immunofluorescence of PФ infected with GFP-*Rv* at 24 h p.i. To obtain better resolution, we captured the images at 100X (in the previous version we did it at 63X) with an optical zoom of 2.9X. It is apparent from the newly added data (Figure 2E) that pATM foci are observed throughout the nuclei. We also observed distinct localization of pATM foci at what appears to be nucleolar. We confirmed the specificity of these signals by performing pATM immunofluorescence in the presence of ATM inhibitor KU55933 (Figure 5B). Treatment of cells with ATM inhibitor led to disappearance of pATM foci, suggesting that the foci indeed correspond to pATM.

Further we made several attempts to co-stain nucleoli with nucleolin (cat no. 396400; Thermo Fisher Scientific) and pATM antibody. Unfortunately, despite our best efforts and even when utilized two other nucleolin antibodies, we were unsuccessful in getting the staining for the nucleolin. Therefore, while we agree with the reviewers that pATM localization in the nucleolus seems to be interesting, we have no definitive experimental proof of this observation.

2) Considering that the formation of γH2AX foci is the strongest evidence for DNA double-strand break induction presented in the manuscript, pictures acquired at higher resolution showing the focal pattern of the staining should be provided. In addition, IF should also be performed using an antibody against another protein forming foci in response to DNA double-strand breaks, for example 53BP1.

Thank you for the comment. We agree with the reviewers that images at higher resolution would provide us a better staining pattern for one of the most important DNA damage marker, γH2AX. Presence of mediator proteins such as 53BP1, which accumulate at the DNA damage site would corroborate our findings. We therefore performed immunofluorescence of PФ infected with GFP-*Rv* at 24 h p.i. for γH2AX and 53BP1 (Figure 2F). To obtain higher resolution images, they were captured at 100X magnification with an optical zoom of 2.9X. Results show distinct focal pattern and colocalization of γH2AX and 53BP1 in *Rv* infected PФ (Figure 2F).

3) When considering (i) the major role of NHEJ in repairing DNA double-strand break in human cells (Ciccia and Elledge, 2010) and (ii) the role of the NHEJ complex DNA-PK in activating Akt (Lan et al. Oncotarget 2016; Stronach et al. Neoplasia 2011), the impact of the inhibition of DNA-PK kinase activity on M. tuberculosis infection should be investigated in this study in parallel to ATM inhibition. This could be tested using one of the specific DNA-PK inhibitors available, such as NU7441.

We thank reviewers for an insightful suggestion. Infection of PФ with *Rv* did indeed result in the activation of DNA-PK, albeit to much lower levels and at later time points compared with the pATM (Figure 6A). To assess the role of DNA-PK and ATR in the survival of pathogen in the host, cells treated with inhibitors individually were infected with *Rv* and CFUs were enumerated (Figure 6C). It is evident that the addition of NU7441 (inhibitor of DNA-PK) or VE-281 (inhibitor of ATR) did not alter the CFUs, suggesting that even though DNA-PK seems to be activated in the DDR pathway, it does not influence the survival of the pathogen (Figure 6C).

4) After treatment with classical DNA double-strand break inducing agent, production of γH2AX is mainly ATM dependent, but has also been reported in some circumstance to be DNA-PK and/or ATR dependent. In Figure 5B, treatment with the ATM inhibitor only reduces γH2AX production, which suggests that other kinases might be at work. The authors should test the implication of ATR and DNA-PK in this residual phosphorylation, since specific inhibitors against these kinases are commercially available.

Results in Figure 5A (Figure 5B in the previous version) showed presence of residual γH2AX despite KU treatment. While phosphorylation of H2AX is predominantly modulated by ATM; ATR and DNA-PKcs may also contribute either partially or entirely to its phosphorylation (Ciccia and Elledge, 2010). Thus, as suggested by reviewers we assessed the possible role of combinatorically inhibiting ATM and DNA-PK or ATR on the levels of γH2AX. PФ treated with KU or KU + NU or KU + VE were infected with *Rv* and the extent of H2AX phosphorylation was evaluated by western blot. While addition of VE+KU did not alter the residual levels of γH2AX, addition of NU +KU resulted in further decrease in the levels of γH2AX, suggesting that DNA-PKcs also contributes to H2AX phosphorylation. To assess the role of DNA-PK and ATR in the ex-vivo survival of pathogen, host cells treated individually with inhibitors were infected with *Rv* and CFUs were enumerated (Figure 6C). It is evident that addition of NU or VE did not alter the CFUs, suggesting that even though DNA-PK seems to be activated in the DDR pathway, it does not influence the survival of the pathogen (Figure 6C).

5) The authors report ATR activation upon infection by the H37Ra attenuated strain. IF should be performed (after H37Ra and H37Rv infection) to monitor for the accumulation of the protein RPA (complex of RPA1, RPA2 and RPA3) in the form of foci, which would support the accumulation of single-strand DNA upon infection (RPA is recruited on single-strand DNA regions).

As suggested by reviewers, we evaluated differential accumulation of RPA2 and pRPA2 in UI, *Ra* and *Rv* infected cells. Infection by *Ra* led to the formation of distinct nuclear pRPA2 (Figure 2C) and RPA2 (Figure 2—figure supplement 1) foci, which corroborate robust activation of ATR-Chk1 pathway (Figure 2B; right panel), suggesting that *Ra* mediates generation of ssDNA. On the other hand, infection by *Rv* showed far fewer foci for RPA2 or pRPA2.

6) Previous work has implicated ROS in the production of DNA damage in infected host cells, the role of ROS should be investigated here. An experiment with N-acetyl-cysteine or membrane permeable reduced glutathione should be performed.

We agree with the reviewers that ROS is known to generate DNA damage as exemplified in case of *C. trachomatis* infection (Chumduri et al., 2013). On the contrary, *H. pylori*, *L. monocytogenes* mediated DSBs are ROS-independent. SodA_Mtb_, which is secreted through SecA2 pathway plays an important role in detoxification of ROS generated inside the host thus partially neutralizing the anti-microbial property of ROS. In *RvΔSecA2* mutant, SodA_Mtb_ secretion is expected to be compromised and hence one would expect higher levels of host ROS upon infection. The fact that *RvΔSecA2* mutant fails to inflict DSBs suggests that *Mtb* mediated damage is unlikely to be ROS-dependent. In agreement with this, addition of N-acetyl cysteine (NAC), a scavenger of ROS, to the infected cells did not alter the γH2AX, suggesting that DNA damage is not mediated through ROS (Figure 7—figure supplement 1B).

7) Figure 6A and B do not provide sufficient data to support conclusions. Rather, the authors should test the effect of the KU inhibitor in vivo.

We agree with the reviewer that in vivo validation of PФ infection data (Figure 7C in the revised manuscript) is necessary. We investigated the possibility of utilizing ATM inhibitor towards adjunct host directed therapy (HDT) for TB with the help of murine infection model. Mice were infected aerosolically with *Rv* and 15 days post infection mice were treated with vehicle or INH or KU or INH + KU for next 15 days and CFUs were enumerated in both lungs and spleen (Figure 7D). As anticipated, treatment with INH resulted in ~ 1 log fold reduction in the load of *Mtb*. While treatment with KU alone did not show any difference in the clearance of the pathogen, combination of KU + INH treatment resulted in ~ 1 log fold better clearance compared with INH treatment alone (Figure 7E). Notably, the impact of KU+INH treatment compared with INH treatment alone was more evident in the spleen (Figure 7F), suggesting that combined therapy compromises dissemination of the pathogen. Taken together, we propose that ATM inhibitor is a potential candidate for HDT.

8) Furthermore, the model suggests AKT signalling promotes survival by blocking M. tuberculosis induced apoptosis. Therefore, M. tuberculosis infection with AKT or ATM inhibition should lead to increased cell death by apoptosis. However, this is very hard to assess with the data presented as, in Figure 5H, the effect of inhibitor alone is not tested (so the increased apoptosis could just be due to the inhibitor – which, as from Figure 5—figure supplement 1, does cause cell death); and in Figure 5—figure supplement 1, there is no increased cell death when M. tuberculosis is added. The authors should assess the effect of the inhibitors (KU and AKT-I) alone on apoptosis without M. tuberculosis infection.

The MTT assay quantifies the relative quantity of viable and healthy cells by measuring the reduction of yellow tetrazolium dye, MTT by mitochondrial enzymes. We agree with the reviewers that inhibitors alone do impact the health of cell, which we could observe when we plotted the graphs at an inhibitor concentration of 10 µM. (Graphs in Author response image 1, same as Figure 6—figure supplement 1 (presently) but plotted UI and *Rv* in the same graph).

**Author response image 1. respfig1:** Cell viability upon treatment with inhibitors.

However, we have not statistically quantitated the difference. To quantify the number of cells undergoing apoptosis we performed Annexin V-7AAD staining wherein Annexin V recognizes the phosphatidylserine (PS) of the inner membrane and 7AAD is a nuclear stain. Even though these two assays utilize different approaches and have different sensitivity levels, in principle we observed similar readout of compromised health of *Rv* infected cells undergoing inhibitor treatment.

To address the point raised by reviewers, we performed Annexin V-7AAD staining both in UI and *Rv* infected cells in the presence of DMSO or KU or Akt-I or KU +Akt-I. Previously we performed the staining at 72 h. Since in MTT assay, we followed only up to 48 h, this time round, we performed Annexin V-7AAD staining at 48 h. While the addition of inhibitors did not impact the percent live cells (Figure 6G; top panel) in UI cells, their presence decreased the percent live cells *Rv* infected cells (Figure 6G; top panel). We observed concomitant increase in the early and late apoptotic cells in the presence of KU or Akt-I or KU + Akt-I (Figure 6G; bottom panel).

9) Western blot data should be quantified and statistically analyzed. Some blots are sliced/edited others not in the same panel (Figure 3C); a single membrane should be used.

To address the concerns, we have quantified and statistically analyzed the blots with major findings. Briefly densitometric analysis of major findings was performed with the help of Image J software. Scans of autoradiograms (300 dpi) were opened in Image J software and the intensities of bands in each lane was individually measured. The intensities of γH2AX/pATM/pATR/pChk2/pChk1/pAkt/p53 were normalized with respect to their corresponding loading control β-actin/β-tubulin/GAPDH. Once the values were obtained for all the lanes in an image, the fold change was calculated as follows.

Fold Change = [Normalized intensity of a band / highest normalized intensity in that panel]

The figure numbers of the corresponding western blot, which is a representative blot of one of the replicates, is given above each graph. The quantitation is provided in the revised manuscript in Figure 6—figure supplement 2 and Figure 7—figure supplement 2.

Sliced blots in Figure 3C have been replaced with unsliced blots. However, the sliced blots in Figure 4B, C, G and H are originally from the same membrane (please find the raw data for Figure 4B, C, G, and H). We have sliced out *Ra* infected lanes from these blots.

**Author response image 2. respfig2:** Raw data for Figure 4B and C. Lanes corresponding to *Ra* (lanes 9-12 in 4B and lanes 4-6 in 4C) have been spliced out. The splice junctions are indicated in the figure.

**Author response image 3. respfig3:** Raw data for Figure 4G. Lanes corresponding to *Ra* (lanes 5 and 6) have been spliced out. The splice junctions are indicated in the figure.

**Author response image 4. respfig4:** Raw data for Figure 4H. Lanes corresponding to *Ra* (lanes 5 and 6) have been spliced out. The splice junctions are indicated in the figure.

10) The model that decreased apoptosis is associated with increased bacterial growth is also contrary to recently published live imaging studies suggesting that macrophage death is essential for rapid bacterial growth (Mahamed et al., 2017). The authors should discuss this.

In the recent paper published in *eLife* (Mahamed et al., 2017), it was reported that internalization of *Mtb* aggregates by the macrophage results in killing of the host cell and there was direct correlation between the size of aggregates and cell death. However, the death in such cases was through non-apoptotic pathway. In our experiments we kept MOI at 1 to circumvent aggregation and non-apoptotic cell death of the host. *Mtb* infection of the host cells can induce host cell death through both apoptotic and non-apoptotic pathways. Apoptotic death is considered to be advantageous for the host as it results in the elimination of pathogen. Induction of non-apoptotic death of the host cells is beneficial for the pathogen as it fails to eliminate the pathogen. The above aspects have been included in the Discussion with appropriate references. We observed significantly increased apoptosis upon the addition of inhibitors for either ATM or Akt or both (Figure 6). These results are in line with the previous findings, wherein bacterial pathogens such as *Salmonella* typhimurium and Coxiella burnetii have been shown to get survival advantage through Akt activation (Kuijl et al., 2007; Voth and Heinzen, 2009).